# Motorway Measurement Campaign to Support R&D Activities in the Field of Automated Driving Technologies

**DOI:** 10.3390/s21062169

**Published:** 2021-03-19

**Authors:** Viktor Tihanyi, Tamás Tettamanti, Mihály Csonthó, Arno Eichberger, Dániel Ficzere, Kálmán Gangel, Leander B. Hörmann, Maria A. Klaffenböck, Christoph Knauder, Patrick Luley, Zoltán Ferenc Magosi, Gábor Magyar, Huba Németh, Jakob Reckenzaun, Viktor Remeli, András Rövid, Matthias Ruether, Selim Solmaz, Zoltán Somogyi, Gábor Soós, Dávid Szántay, Tamás Attila Tomaschek, Pál Varga, Zsolt Vincze, Christoph Wellershaus, Zsolt Szalay

**Affiliations:** 1Department of Automotive Technologies, Budapest University of Technology and Economics, Műegyetem rkp. 3, 1111 Budapest, Hungary; tihanyi.viktor@kjk.bme.hu (V.T.); mihaly.csontho@auto.bme.hu (M.C.); viktor.remeli@auto.bme.hu (V.R.); andras.rovid@auto.bme.hu (A.R.); zsolt.vincze@auto.bme.hu (Z.V.); zsolt.szalay@auto.bme.hu (Z.S.); 2Department of Control for Transportation and Vehicle Systems, Budapest University of Technology and Economics, Műegyetem rkp. 3, 1111 Budapest, Hungary; 3Institute of Automotive Engineering, TU Graz, Inffeldgasse 11, 8010 Graz, Austria; arno.eichberger@tugraz.at (A.E.); zoltan.magosi@tugraz.at (Z.F.M.); christoph.wellershaus@tugraz.at (C.W.); 4Department of Telecommunications and Media Informatics, Budapest University of Technology and Economics, Műegyetem rkp. 3, 1111 Budapest, Hungary; ficzere@tmit.bme.hu (D.F.); magyar@tmit.bme.hu (G.M.); soos@tmit.bme.hu (G.S.); pvarga@tmit.bme.hu (P.V.); 5Automotive Proving Ground Zala Ltd., Fészek u. 4, 8900 Zalaegerszeg, Hungary; kalman.gangel@zalazone.hu; 6Linz Center of Mechatronics GmbH, Altenberger Straße 69, 4040 Linz, Austria; leander.hoermann@lcm.at (L.B.H.); maria.klaffenboeck@lcm.at (M.A.K.); 7ALP.Lab GmbH, Inffeldgasse 25F/5, 8010 Graz, Austria; christoph.knauder@alp-lab.at; 8JOANNEUM RESEARCH Forschungsgesellschaft mbH, Steyrergasse 17, 8010 Graz, Austria; patrick.luley@joanneum.at (P.L.); matthias.ruether@joanneum.at (M.R.); 9Knorr-Bremse R&D Center Budapest, Major u. 69, 1119 Budapest, Hungary; huba.nemeth@knorr-bremse.com; 10Virtual Vehicle Research GmbH, Inffeldgasse 21a, 8010 Graz, Austria; jakob.reckenzaun@v2c2.at (J.R.); selim.solmaz@v2c2.at (S.S.); 11Department of Electrified Powertrain, Transmission & Software, AVL Hungary, Irinyi József u. 4-20, Science Park, 1117 Budapest, Hungary; zoltan.somogyi@avl.com; 12Department of GIS, Budapest Public Road, Bánk bán u. 8-12, 1115 Budapest, Hungary; david.szantay@budapestkozut.hu; 13ITS Department, Hungarian Public Roads, Fényes Elek u. 7-13, 1024 Budapest, Hungary; tomaschek.tamas@kozut.hu

**Keywords:** vehicle detection, automated driving, autonomous vehicles, measurement campaign, 5G, vehicle sensors, infrastructure sensors, UHD map

## Abstract

A spectacular measurement campaign was carried out on a real-world motorway stretch of Hungary with the participation of international industrial and academic partners. The measurement resulted in vehicle based and infrastructure based sensor data that will be extremely useful for future automotive R&D activities due to the available ground truth for static and dynamic content. The aim of the measurement campaign was twofold. On the one hand, road geometry was mapped with high precision in order to build Ultra High Definition (UHD) map of the test road. On the other hand, the vehicles—equipped with differential Global Navigation Satellite Systems (GNSS) for ground truth localization—carried out special test scenarios while collecting detailed data using different sensors. All of the test runs were recorded by both vehicles and infrastructure. The paper also showcases application examples to demonstrate the viability of the collected data having access to the ground truth labeling. This data set may support a large variety of solutions, for the test and validation of different kinds of approaches and techniques. As a complementary task, the available 5G network was monitored and tested under different radio conditions to investigate the latency results for different measurement scenarios. A part of the measured data has been shared openly, such that interested automotive and academic parties may use it for their own purposes.

## 1. Introduction

Autonomous vehicle technology is the main determinant of the automotive industry in our days. Highly automated and soon fully autonomous vehicles will revolutionize road transportation [1,2]. Although this trend is very spectacular, in the background a vast amount of engineering research and development (R&D) work is required, i.e., Connected Automated Vehicles (CAVs) and all related technologies require complex testing and validation procedures [3,4,5,6].

A significant part of these R&D tasks needs proper test measurement data concerning the vehicle dynamics together with proper map data (i.e., the punctual environment information around the vehicle under test). Additionally, the monitoring of vehicles is of crucial importance in this technology, e.g., IoT (Internet of Things) sensors are increasingly used [7,8]. However, collecting such data is not straightforward and it takes a great deal of time and money. It is a general problem that high precision map data are not available for automotive tests. Moreover, the sensing of the car and related processing/fusing algorithms cannot be validated against real-world data, as, usually, during the test measurement, there is no other external detection that records the car’s motion on HD map [9,10]. Thus, it is a real problem to measure and validate the performance of a sensor fusion algorithm under real road conditions.

As an effective solution to this challenge, a grandiose measurement campaign was carried out on a real-world motorway under the umbrella of international cooperation. The paper presents the process and the main achievement of this measurement campaign. When using the terminology of a recent V2X testing survey [11], our paper covers V2X performance testing (especially latency testing) that involves vehicle gateway testing. Nevertheless, the major achievement here is presenting how live field-testing has been planned and executed in order to evoke further, digital-twin-based validations through Vehicle-in-the-Loop (ViL) [5,12] or Scenario-in-the-Loop (SciL) [13] methodologies. The concepts and perspectives of ViL and SciL have been summarized well by [14].

Communication between autonomous vehicles and between vehicles and the environment further helps in smoothing traffic and improving safety. Out of the various technological candidates for such Vehicle-to-Everything communication (V2X), mostly two can be considered to be top competitors: dedicated short range communications (DSRC) [15] and cellular mobile-based technologies, especially 5G [16], which is expected to be used for various URLLC (ultra-reliable low latency communications) applications [17]. The interworking possibilities of DSRC and 4G+ (LTE advanced) cellular technologies are surveyed in [18]. In the current paper, we utilized a 5G NSA (Non-Standalone Architecture) infrastructure that was dedicated for our measurement campaign.

In 2018, Hungary and Austria signed a cooperation agreement on the development and testing of electric, connected, and self-driving automotive technologies. Subsequently, at their meeting in Vienna in February 2019, the Hungarian Minister of Innovation and Technology and the Austrian Minister of Transport agreed to launch a bilateral R&D cooperation to support the preparation of Austro-Hungarian R&D and innovation projects. In relation to this cross-border cooperation, four R&D projects have been defined (see the Acknowledgment section). As an important basis to support these projects, a measurement campaign has been designed and successfully fulfilled. The campaign was especially important for the R&D project, called “Development of central system architecture for autonomous vehicle testing and operation related services (2019-2.1.14-ÖNVEZETŐ-2019-00001)”, where the core is developed and tested based on the achieved measurement data (see the project goal summarized in a picture in Figure 1).

The applied test field was assigned on a Hungarian motorway stretch. To securely conduct the measurement activity, the test field was fully closed to traffic during the campaign, i.e., the test vehicles had access to this route solely. The goal of the measurement campaign, as presented in the paper, was to realize fully comparable measurements (i.e., data that were detected by vehicles and by sensors located in the infrastructure) being essential to the implementation of the technologies to be developed in later phases of automotive R&D projects. Meanwhile, the communication architecture (5G cellular network) was continuously tested on vehicles and infrastructure both in terms of latency and data transmission with ex-post evaluation.

The participants of the presented measurement campaign are listed below:Budapest University of Technology and EconomicsGraz University of TechnologyALP.Lab GmbHAutomotive Proving Ground Zala Ltd.Hungarian Public RoadsJOANNEUM RESEARCH Forschungsgesellschaft mbH.Knorr-Bremse HungaryAVL HungaryBudapest Road AuthorityVirtual Vehicle Research GmbHLinz Center of Mechatronics GmbH

The main contribution of our work is that the data collected are suitable for the validation of environment perception algorithms and dependent functions for specific intelligent vehicle setups. Concerning the related works, although there exist freely available test data sets, they are not always applicable for validation, since they might not contain the exact sensor setup that we wish to test and the distribution shift might be too large to neglect. A well-known automotive dataset is KITTI [19], where ground truth labeling was performed manually, and the distribution of the collected data is highly setup specific. An example for a different kind of dataset is NGSIM [20], which determines ground truth via infrastructure sensors solely, and does not assume any particular sensor setup. Clearly, it makes sense to automate ground truth generation if new datasets have to be recorded on a per-setup basis; our measurement campaign is a proof-of-concept demonstration of exactly such a method. A thorough search of the relevant literature only yielded one similar article [21], which also describes a measurement campaign, the goal of which, however, is different, i.e. it focused on localization issues in GNSS-denied/challenged environments and not on automotive problems. In order to demonstrate the paper’s contribution, a part of our measurements is shared as open access data for R&D activities (https://automateddrive.bme.hu/downloads (accessed on 19 March 2021)).

Traditionally collected automotive data sets differ from our approach in two major ways: (1) sensor data are collected by a single party and (2) ground truth information is acquired via manual labeling. Therefore, conventionally collected data sets are usually constrained by two major drawbacks: they prohibit scaling to crowdsourced data collection schemes, and the reference labeling might suffer from quality issues, especially in cases where exact 3D object position and orientation are to be determined. Conversely, our approach is a first step towards collaborative data collection, and we perform our measurements in a highly controlled environment allowing for us to exclusively utilize high-precision automatic data labeling.

The Providentia team [22] took a different labeling approach in 2019, where a form of crowdsourcing was achieved by using a helicopter that was equipped with a 4k camera system and a high-end MEMS GNSS/IMU for birds eye view ground truth generation. The advantage is that even non-intelligent and non-affiliated vehicles could take part in the measurement, therefore the setup of a controlled environment with road closures was not necessary. The disadvantage is the high cost of the equipment and/or the possibly decreased ground truth precision, since relying on very precise flying localization and deep learning based image processing can introduce its own specific source of error. Shan et al. [23] obtain ground truth on pedestrians while using RTK receiver units. An in-depth treatise of RTK based ground truth generation for SLAM evaluation is given in [24], while taking uncertainties and their effect on the evaluation into account. A framework for cost-effective manual improvement of GPS ground truth quality is described by [25].

The paper’s structure is as follows. Section 2 presents the measurement campaign realization introducing the test field, the vehicle, and the infrastructure sensors. In Section 3, the 5G technology based measurement technology is discussed. Section 4 summarizes the measurement results of the LIDAR, camera, radar, and GNSS devices, showcasing two application examples. The paper ends with a brief conclusion.

## 2. Measurement Campaign Realization

In this section, the measurement campaign realization is presented by introducing the test field, the HD map, the vehicle, and the infrastructure sensors in detail. First the specifics of the measurement campaign location are given, including a detailed description of the test site sections. Subsequently, the HD mapping process is described by both Joanneum Research and Budapest Road Authorities, specifying the measurement equipment used and the characteristics of the resulting maps. Next, the gantry infrastructure sensors that were installed by BME and overlooking the test road sections are discussed. Following this, the participating test vehicles are described in sequence by following partners: ALP.Lab, BME, Knorr-Bremse, AVL, Virtual Vehicle, Joanneum Research, TU Graz, and, finally, the Linz Center of Mechatronics, whose sensors were mounted onto the Virtual Vehicle’s platform.

### 2.1. Location of the Measurement Campaign

The measurement campaign was carried out on a highway section near the town of Csorna, in the North-Western part of Hungary (Győr-Moson-Sopron county, Western Transdanubia Region). Csorna is located in the crossing of two main regional highway sections, one is the M85 running east-west, connecting the Fertő-Hanság region to Győr, and via M1 to the Capital City of Budapest, and the other way to Burgenland in Austria, and via A3 to Vienna. The other one is the M86 running from north to south, is part of the TEN-T comprehensive network, and part of the E65 route running from Sweden (Malmö) to Greece (Crete) at the same time. Both of the roads are part of the Hungarian State Public Road Network, and they are managed by the national road operator company, Hungarian Public Roads.

The two highways have the same route for a short section, which is the eastern bypass section of Csorna, and, after this joint section, the M85 turns west and proceeds to Austria, and M86 heads south towards Szombathely. M86 is in service between Szombathely and Csorna only, the extension is planned to both north and south. Currently, the route ends at northern end point of the Csorna bypass section, and leads back to the main road no. 86 a couple of kilometers after the departure of M85 route. This is a transition section where the highway cross section turns into a 2 × 1 lane road, and it ends in a roundabout with main road 86. The measurements took place at this transition section, at the temporary end junction of M86 (Figure 2). From south to the north, there are four sections with different characteristics:Section 1. Interchange area (red): the two carriageways have different horizontal and vertical alignment, while leaving the M85-M86 interchange. In this section, two 3.50 m wide lanes are available for the through traffic, and there are additional accelerating/decelerating lanes linked to junction ramps.Section 2. Open highway (blue): a common approx. 300-m-long dual carriageway section with two 3.50 m wide traffic lanes, and 3.00 m wide hard shoulder on both sides.Section 3. Transition section (yellow): heading towards north, the central preserve ends, and only one 3.50 m wide lane left for each direction, while northbound lane is diverted to the eastbound carriageway.Section 4. Main road (purple): the last section is a single carriageway road with 2 × 1 lanes, connecting the highway section to main road number 86.

Two measurements were run parallel using mainly sections 1 and 2 of both (East and West) carriageways at the same time.

Besides the physical infrastructure, a high bandwidth fiber optic communication network is available along the entire section, and several sensors, cameras, and VMS gantries were deployed with fixed power supply. Using this infrastructure, additional sensors (cameras, laser scanners, etc.) and mobile 5G base stations were deployed during the tests.

### 2.2. High Precision Mapping during the Measurement Campaign

Joanneum Research and Budapest Road Authority both carried out high precision mapping during the measurement campaign. The two datasets were collected while using different instruments: Leica Pegasus 2 mobile mapping system and RIEGL laser scanning system. An additional advantage of this doubled measurement is that the datasets about the same test field can be later further analyzed and compared.

### 2.3. Ultra High Definition Mapping for Automated Driving

Ultra High Definition maps (UHDmaps) can also be seen as digital twins of reality, including every detail of test environments at the highest accuracy. Those maps were specially made to serve as a quality proofed source of reference for testing and validation of ADAS/AD driving functions with an absolute accuracy of +/−2 cm. In comparison, commercial high definition maps provided by map companies, like HERE or TOMTOM, are focused on supporting driving functions with an absolute accuracy of +/−20 cm only [26], which is to little for test and validation purposes. The digital twin includes the road surface and any traffic infrastructure, as well as the road topology and environment models (topography). Basically, every object in the environment that could or should be sensed and recognized by the environmental perception system of an automated car has to be included in the digital twin at the highest detail, thus serving as a source of reference to evaluate the performance of automated vehicles. Basic examples are the recognition of traffic signs or lane borders that are needed for basic driving functions, like lane keeping and adaptive cruise control. Any object recognizable by vehicles can be validated by comparing the recognition results with the digital twin, which serves as a quality proven source of static ground truth. Currently the creation of digital twins of test tracks or proving grounds is a very challenging and resource intensive task. This especially applies in the scope of testing automated vehicles, where highest accuracy and detail is needed. Basically, a digital twin is created from survey data, which are captured by mobile mapping systems, including high resolution cameras and laser scanners. Such kind of mobile mapping devices are normally designed as survey-grade stand-alone capturing systems which can be used with any vehicle by mounting the system on the roof. After complex post-processing of the data, geo-referenced measurement images are retrieved (every pixel has a known absolute location on earth in centimeter accuracy) and very dense and accurate LIDAR (Light Detection and Ranging) point clouds representing a 3D scan of the environment (each point has a centimeter-accurate absolute 3D location and a reflection intensity value) are available. These data are the basis for deducing the digital twin or ultra high definition map, which is currently carried out in a very complex and mainly manual workflow by the use of expert software with semiautomatic tools. The creation of digital twins, including: (i) the survey with high class mobile mapping systems and (ii) the semantic processing of the data is very time consuming and costly. In order to comply to the European general data protection regulation, any image data have to be anonymized as early as possible. Personal data, like faces or number plates, are removed by automated tools, after that, a manual quality check is performed. Any labeling of objects should be done by the use of already anonymized data. The final digital twin includes a digital representation, including road surfaces and topology, as well as 3D models of any relevant object, like bridges, guard rails, traffic signs, etc. Finally, in order to make the data usable within several simulation and evaluation tools to test automated vehicles, the digital twin has to be exported to different data formats that are suitable for either visualizing or comparing the data, or using it directly in simulation and rendering engines. THe typical standards used in these cases are ASAM OpenDrive/OpenCRG, IPG road5 (Simulation in Vires VTD or IPG Carmaker), Filmbox FBX (3D Display in Blender, Unity or Unreal Engine), GeoJSON, or Shapefiles (Measurements in GIS Software).

#### 2.3.1. UHD Mapping by Joanneum Research

Figure 3 shows the state-of-the-art of digital twin (UHDmaps) production, which is a first approach by Joanneum Research. The automated creation is still subject of research and it is focusing on the development of robust automated production workflows by the use of artificial intelligence. Automation will enable cost efficient creation and frequent updates of digital twins and it will pave the way for digital twin based test and evaluation of automated vehicles and their environmental perception systems.

The depicted semi automatic production workflow was applied for producing UHD maps for the M86 highway section. In order to to provide UHDmaps in time, a mobile mapping survey of the road section was carried out with a Joanneum Research measurement vehicle using a Leica Pegasus 2 Ultimate Dual Head mobile mapping system (that was funded by “Zukunftsfond Steiermark”, see Figure 4) in advance to the measurement campaign.

Based on the survey data (left of Figure 5), UHDmaps (right of Figure 5) have been deduced through a semiautomatic workflow. This was achieved by using object detection and AI backed classification methods. The automatically deduced basic map features include: (i) lane borders and markings, (ii) lane center lines, (iii) curbs and barriers, (iv) traffic signs and light poles, and (v) road markings. Additionally, semantic meta data, like the road topology (lane numbers, traffic rules, etc.) and sign types, marking types, or sign texts are also embedded. As a final step, the UHDmap data were exported to the ASAM OpenDrive simulation format as well as to KML and GeoJSON.

Finally, a human quality control was carried out to ensure the completeness and correctness of the created UHDmap, which served as a quality proven source of static ground truth for the complete measurement campaign.

#### 2.3.2. HD Mapping by Budapest Road Authority

During the Road Information System (RODIS) project, which was owned by Budapest Public Road (Road Management Co. of Municipality of Budapest, Hungary) the company introduced the 3D mapping services and technologies. For reaching these aims, the development started in 2013, with a 3D laser scanning and modeling of Budapest. So far, the RODIS team has already scanned 15,000 km and processed the scan data in survey grade accuracy and also extracted 2.5 million 3D features from mobile and terrestrial point clouds for engineering and asset management. The Budapest Road Authority joined the M86 measurement campaign to utilize this kind of experience for HD maps that was earned by mapping the road network of Budapest. During the measurement campaign, the data capturing was performed by a Riegl VMX-450 mobile laser scanner system (with a Flir Ladybug 5+ camera system), which offers extremely high measurement rates, providing dense, accurate, and feature-rich data, even at high driving speeds. The roof-carrier mounted measuring head integrates two laser scanners as well as inertial measurement and GNSS equipment. Fast 3D data collection, featuring high accuracy and high resolution, provides a basis for producing HD map. After the data capturing, it is needed to do the post-processing of the data. During the post-processing phase within the project-oriented software environment, it is possible to manage all of the data acquired and processed within a single project. These data include project data, scanning system information data, such as mounting information and calibration data, laser raw data, e.g., the digitized echo signals from the RIEGL laser scanner, position, and orientation data from the IMU/GNSS system, intermediate data files, and georeferenced point cloud data with additional descriptors for every measured coordinate. At the post-processing of the data, ground control points (that were measured during the M86 measurement campaign) were also imported. The ground control points measurement procedure is one of the most important elements for the creation of point cloud dataset, determining the geometric accuracy and the quality of the final product. For the HD map, it is necessary to produce map content with pre-defined object. It is useful to design a database before the data extraction as well. Budapest Road Authority chose the geodatabase (.gdb) format, which is an ideal solution that is designed to store, query, and manipulate geographic information and spatial data (such as vectors). Filling the geodatabase, the data extraction was processed from the 3D classified point clouds manually or semi-automatically (Figure 6 and Figure 7).

The quality control of the data was carried out automatically and by humans as well to meet the highest standards. The adequate quality control for the standards is also inevitable; otherwise, it would not be possible to import the data of the geodatabase into OpenDRIVE format (which physical map format in self-driving was chosen for this project).

After the M86 measuring campaign, it became clear that no one will solve the problem of generating HD map on the level of the accepted quality and quantity alone. The only possible way is to work with partners who have outstanding knowledge and experience in their respective fields. For producing the HD maps, the close partnerships are a very effective way to step forward.

#### 2.3.3. Comparison of the Datasets Collected by Leica and RIEGL Laser Scanning Systems

For the measurement campaign, two mobile mapping systems were used: a Riegl VMX-450 provided by Budapest Road Authority and a Leica Pegasus 2 Ultimate provided by Joanneum Research. It was sought to compensate for shortcomings from either survey system: the Ladybug 5+ on the Riegl provides a better dynamic range and produces higher resolution images than the integrated 360${}^{\circ }$ camera on the Leica. The combined resolution of the Riegl Image yields 36 mega pixels, whereas the Leica Image has its resolution boundary at 20 MP. Preferring the Ladybug images is particularly beneficial when identifying small details, such as traffic supplemental signs (Figure 8).

On the other hand, the Z+F laser scanners on the Leica Pegasus 2 Ultimate Dual Head offer twice the point sample rate (1 Mio points) at the same rotation speed (200 Hz) than the VQ-450 Scanners on the Riegl. Both of the systems are dual profiler setups, so the total point sampling rate has to be doubled. This is relevant, for example, when creating a surface model of the ground or identifying exact boundaries of lane markings (Figure 9). When comparing the ground level scan pattern of both systems, the better point density of the Leica system is demonstrated by a denser fishnet shape. Thus, surface properties, such as slope and lateral profile, as well as cracks or holes, can be identified faster.

Note that this paper intended to show the differences between two different commercial mobile mapping systems, concerning their data quality (LIDAR, imagery), and did not aim to address how image data quality could be optimized in post-processing [27]. Increasing image quality is a part of future work.

### 2.4. Infrastructure Sensors of the Team from Budapest University of Technology and Economics (BME)

Intelligent infrastructure is a rapidly evolving area with certain test systems achieving real-time data fusion capabilities, in both centralized [22] and decentralized I2V setups [23]. Some even experimented with cloud-based control [28]. BME is developing an edge cloud based connected infrastructure and vehicle system that, with precise calibration and time synchronization, allows for complete and real-time integration of sensory data of all participants (not only the infrastructure). In the M86 measurement campaign, we demonstrate the offline capabilities of such a system.

The measurement campaign’s closed-off highway section contains several gantries that were equipped with variable message signs and various sensors. On the section between the overpass and the M85 junction, the two gantries closest to the overpass were used by the BME team as infrastructure installation points for overhead sensors that became the static data collection nodes in each measurement scenario (Figure 10). The two gantries lie across different sides of the road and at a longitudinal distance of approximately 160 m from each other.

Installation was made possible by the tight collaboration with Hungarian Public Roads having on-site, remotely controlled cameras that provided online feedback on the overall situation.

BME deployed two measurement stations, one on each of the two gantries; the one closer to the M85 junction referred to as Infra1 and the one closer to the overpass as Infra2. The stations were installed directly above the innermost barrage line marking of the road, thus having a good overview of both highway directions. Both of the stations were equipped with one top-down mounted LIDAR and two cameras pointed downwards at an angle, looking in both directions. The two stations were synchronized with each other and the rest of the system via a GPS based NTP (Network Time Protocol (NTP)) timeserver provided by Cohda MK5 OBU-s. Both of the stations contained a measurement control PC that ran RTMaps sensor data processing framework for data collection and it was accessed remotely over 4G, allowing manual control and monitoring of the ongoing measurements from within the central tent.

The sensor setup was the following: Infra1 station used one Ouster OS1 64-channel LIDAR and two Hikvision DS-2CD2043G0-I $103^{\circ }$ FOV ingress protected (IP) cameras with IR illuminator for nighttime imaging. Infra2 station used one Ouster OS1 64-channel LIDAR and two Sekonix SF3325-100 2.3 megapixel $60^{\circ }$ FOV cameras with FAKRA Z type connectors, necessitating data collection via a specialized measurement PC (Nvidia DrivePX2 computing platform). The LIDAR-s provided 3D pointclouds with intensity levels over Ethernet at 10 Hz. The Hikvision cameras produced H.264 compressed video streams over Ethernet, while the Sekonix cameras provided raw image sequences at 30 Hz. An example image taken from Infra2 station is shown by Figure 11.

### 2.5. Vehicles and Vehicle Sensors

In the following parts, the applied vehicles and vehicle sensors of all partners are introduced.

#### 2.5.1. ALP.Lab Test Vehicle and Vehicle Sensors

The Austrian Light-Vehicle Proving Region for automated driving (ALP.Lab) again attended [29] a joint measurement campaign in the region around ZalaZone. The conducted highway measurements were supported using one ALP.Lab test vehicle that was equipped with a high accuracy DGPS (Differential Global Positioning System) vehicle position tracking unit. The vehicle that was used for the testing activities was a modern Mercedes A-class from model year 2018 (see Figure 12). A NovAtel PwrPak7 GPS unit was used for accurate position tracking during the measurements, while the measurement of vehicle motion is performed by an Oxford Technical Solutions (OxTS) RT3000v2 Dual antenna IMU-System. The IMU-System used can achieve 1cm RTK (Real Time Kinematic) accuracy in both functions of real-time and post-processing. The measurement outputs are time-stamped, referred to GPS time, and the inertial measurements are subsequently synchronized to the GPS clock. Measurements have been conducted using different pre-defined driving scenarios of a vehicle group. The collected data of the vehicle trajectories with the DGPS systems are used as a basis for ground truth related data processing and evaluation.

#### 2.5.2. BME Measurement Vehicle Sensors

During the M86 measurement campaign, the infrastructure as well as the participating measurement vehicles were equipped with multiple sensors of different types, which enabled us to collect real-time data of the surroundings as well as the locations of all participating vehicles. These data serve as reference values that are useful for testing and evaluating perception related methods and algorithms from different aspects. Let us give a brief overview on sensor setup of the perception vehicle that was developed by the BME and used during the M86 measurement campaign. The measurement vehicle is equipped with seven 2Mpixel cameras having 60 and 120 degrees FOVs, two tilted side LIDARs, each having 16 channels and a single 64 channel LIDAR mounted in the middle of the rooftop of the vehicle (see Figure 12). Each LIDAR was configured to run at 20 Hz, which was their maximum achievable framerate. In addition to cameras and LIDARs, there is differential GPS with a dual antenna system installed near the center of mass of the car, also including an IMU that was configured to run at 100 Hz. To collect and process the incoming data stream generated by the sensors in real-time, an NVidia DrivePX2 platform was deployed together with a software framework to execute multi-sensor applications. For data transfer, there is a V2X on board unit installed together with a 5G router. Besides the individual sensors, another important aspect was their synchronization, which was based on the GPS time and the NTP. Because each participating vehicle and the infrastructure relied on the GPS time as reference time, all of the collected data could be kept synchronized during the whole measurement campaign, which is one of the most crucial parts of the system. The NTP service was configured to run on the V2X (Vehicle To Everything) platform as well as on the Nvidia DrivePX2. In the multisensor software framework to each incoming data frame, a timestamp—obtained through NTP—was assigned, based on which an accurate time synchronization could be achieved. For the calibration of cameras, a flexible chessboard based technique has been used [30] while the estimation of camera-LIDAR extrinsics was performed based on the method shown in [31].

#### 2.5.3. Knorr-Bremse Vehicles and Sensors

Knorr–Bremse participated in the project with two test vehicles, a MAN TGX heavy duty tractor in combination with a Krone semitrailer and an MAN TGA heavy duty solo tractor (see Figure 13).

Both of the test vehicles have been equipped with a highly automated driving system and a corresponding environment perception setup. A detailed description of the local situation awareness system is found in [32].

Figure 14 depicts the environment sensor setup. The tractors have redundant sensor pairs for most of their sensors and corresponding field-of-views. Both of the test vehicles have been equipped with the same environment sensors.

During the measurement campaign, the following sensors have been used. The front zone of the vehicles are covered by a pair of forward looking cameras (6 and 6${}^{\prime }$), a pair of forward looking long range radars (3 and 3${}^{\prime }$), and a forward looking wide angle short range radar (12). The side zones of the tractors and the combination are covered at each side by wide angle short range radars (4 and 4${}^{\prime }$) and front corner placed down looking fisheye cameras (8 and 8${}^{\prime }$), where both of the fisheye cameras cover the vehicle front and the corresponding vehicle sides. The rear zone is covered at each side by rear looking cameras (7 and 7${}^{\prime }$) and rear looking long range radars (5 and 5${}^{\prime }$), each being installed to the mirror consoles of the vehicle and, finally, a rear looking long range radar (10), which is installed to the middle of the rear axle. If a trailer is connected, the rear middle long range radar (10) is looking out underneath the trailer to the ego lane. Besides environment perception sensors, both test vehicles have also been equipped with dGNSS and IMU sensors.

#### 2.5.4. AVL Vehicles and Sensors

AVL Hungary participated in the measurement campaign in cooperation with AVL Software and Functions Regensburg. The purpose of the campaign was to record special use cases and reliable environment data with a high definition sensor setup. An Audi A4 was the test vehicle for this measurement campaign, which was equipped with Dynamic Ground Truth (DGT) rooftop box (Figure 15). The recorded data will be used for offline validation of a self-developed AI based perception software. Moreover, advanced driver assistant systems can be tested with real-world scenarios in simulation.

The rooftop box contains several high resolution and automotive certified sensors. There are four cameras giving a 360${}^{\circ }$ field of view around the vehicle. The DGT has three lidars in total, one main lidar with 200 m range and two looking at the sides with 100 m range. The accurate position of the DGT and the vehicle are measured by a differential GNSS unit with built-in RTK functionality. The synchronization of the data streams is handled by an industrial PC inside the rooftop box. The uncompressed raw data are stored in generic Robot Operating System (ROS) rosbags.

#### 2.5.5. Virtual Vehicle AD Demonstrator and Sensors

Virtual Vehicle Research GmbH (VIF) is a research organization that is actively working on all areas of model-based vehicle development, particularly including automated driving system solutions. Of special interest is the development of tools and methodologies that can aid in the scenario based validation and verification of ADAS systems at various abstraction levels spanning simulation-only and real-life testing [33,34]. With this motivation and background, VIF joined the M86 measurement campaign with one of its generic Automated Drive Demonstrator (ADD) vehicles. A Ford Fusion MY2018 was the vehicle used for this purpose, which is equipped with several additional sensors and computational hardware as well as custom software components. Figure 16 shows the utilized sensor equipment.

The ADD sensor setup can be modified, depending on the measurement or the use-case requirements. To support the aim of this measurement campaign, the ADD vehicle was equipped with a high-accuracy multi-antenna DGPS system to provide a ground truth information. For this purpose, a Novatel ProPak6 RTK-GPS system was utilized for the measurement of the precise position of the vehicle with respect to the acquisition of the desired dynamic ground truth. Additionally, the VIF ADD vehicle also logged other sensor data, in parallel to the GPS, which is relevant to the perception algorithms, specifically including a Continental ARS408 long-range RADAR sensor and an Ouster OS1-64 LIDAR sensor. Figure 16 shows the mounting positions of the perception sensors, where the RADAR sensor is marked in green and the LIDAR is marked in blue. For the data acquisition, ROS-based middleware AUTOWARE running on an Ubuntu X86 PC mounted in the trunk of the ADD was utilized.

#### 2.5.6. Joanneum Research Vehicle and Sensors

The Dynamic Ground Truth (DGT) cloaked sensor system of Joanneum Research puts special emphasis on image quality combined with high-precision localization in an unobtrusive housing. The sensors are rooftop-mounted and covered by a standard ski-box (Figure 17). Four 12 Megapixel cameras provide a seamless surround-view, combined with 360 degree LIDAR coverage in 32 scanning planes. A dual-antenna system, combined with either a MEMS IMU from Novatel or a fiber-optic IMU from IMAR, provided robust and highly precise location information. The system can optionally be extended by a forward-looking stereo system and a Radar-System. The Radar can be operated in SAR-mode for static object mapping and localization, or in the forward-looking object detection mode. All of the components are hardware-synchronized to provide optimal concurrency of camera, LIDAR, and Radar readings. A three-CPU PC system with two GPUs provides sufficient power for real-time data analysis and storage, at a power level that is suitable for vehicle battery operation.

#### 2.5.7. TU Graz Vehicles and Sensors

The Institute of Automotive Engineering at the Graz University of Technology (TU Graz) took part in the M86 measurement campaign with two vehicles. One vehicle has been configured as the host vehicle (BMW640i), while the other car (Seat Leon) has been configured as the target vehicle, see Figure 18. The host vehicle was additionally equipped with various environment recognition sensors with an accessible data interface, parallel to those that are used in series vehicles for the state of the art ADAS functions.

The experimental set-up was developed in previous projects and it has proven to have suitable performance for ADAS/AD testing with respect to the robustness and usability while testing, the accuracy of the reference measurement system, and the time synchronicity between reference measurement and perception sensor output [35]. The data derived in several thousands of on-road testing were used for the calibration of different phenomenological perception sensor models [36,37,38].

In order to receive as much information as possible from the vehicle’s surroundings and from other road users, the host vehicle was equipped with two Continental Long Range RADAR sensors, one for the target-, one for object recognition, as well as a MobilEye Front Camera Module, with a Robosense Laser scanner and the Cohda MK4 C2C communication platform. In addition to the environment recognition sensors, both of the vehicles were equipped with a special measuring system. The DEWETRON-CAPS measuring unit uses, in each car, the GeneSys Automotive Dynamic Motion Analyzer (ADMA) inertial measuring system for motion analysis with six degrees of freedom in combination with the powerful NovaTel RTK-GPS receiver to provide highly accurate Ground Truth data of the vehicles, see Figure 19.

The relative position of the target vehicle to the host vehicle is calculated online in the host vehicle during the measurement, which receives the position through the data transfer of the target vehicle via a special WiFi Link, which was designed for automotive applications. Beside the ADAS related environment recognition sensors, a video camera was also connected to the data acquisition unit in both vehicles to record GPS-time synchronized video data from each driving scenario. The above-described setup makes it possible to control the ongoing driving scenario from the host vehicle online by verifying the real time available position and dynamic information of the target vehicle, and then record the output data of all additional mounted perception sensors and signals of vehicle bus system of both vehicles that were synchronized with the GPS-time.

#### 2.5.8. Linz Center of Mechatronics GmbH Sensors

The Linz Center of Mechatronics GmbH participated as a representative of the FleetQuAD exploratory project, which targets an evaluation of different use cases that were related to the evaluation of the road and road infrastructure itself regarding the capability of ADAS/AD. The main goal of this measurement was the evaluation of accelerometer and gyroscope data with respect to the unevenness of the road, beginning with corrugations in road surface over cracks ending with potholes; similar approaches were presented in [39,40]. The measurement setup consists of two XSens MTi670 for recording acceleration at 1000Hz and gyroscope values at 800Hz. It also includes a GPS receiver, which acts as high precision synchronization unit using the received time information, as well as location information source. Location and time-synchronizing the data are essential to correlate the recorded measurement data to a specific road segment. The sensors were mounted in the rear trunk of the VIF ADD vehicle that is shown in Figure 16, with the *z*-axis being parallel to the gravity vector and the *x*-axis parallel to the driving axle of the vehicle.

An exemplary analysis of accelerometer data is shown in Figure 20. The spectrograms show the difference of an even road (newly made road surface) with a single disruption at about 2s in contrast to the evaluated pattern of a “normal” road (intact, but used), which shows significantly more oscillations at approximately the same vehicle speed of $10.4\;\frac{m}{s}$ and using the same sensor unit. All of the speed data are estimated based on the available GPS data. At the bottom of Figure 20, the time signals of two different accelerometer measurements are compared. They are in good accordance at the peak at 2s. Even the according GPS positions (that are marked with crosses in the right plot) are in good accordance, which makes the educated guess that the origin was the identical disruption, even if the velocities were slightly different with approximately $11.3\;\frac{m}{s}$ and $12.95\;\frac{m}{s}$.

Analogously, the data of the gyroscope can be processed, although the accelerometer shows better conformity among the different measurements. A note aside: the road along the test route was perfectly prepared, such that it was hard to detect congruent disruptions.

## 3. 5G Technology Based Measurement Results

### 3.1. 5G V2X State of the Art

Vehicular communication is a critical technology for providing connectivity between vehicles, roadside units, and pedestrians in the intelligent transportation system. There are several wireless technologies for automotive network connectivity, such as traditional Wi-Fi, IEEE 802.11p, and cellular communications. Cellular V2X (C-V2X) was recently developed by the third-generation partnership project for automotive facilities (3GPP). 5G can be the primary enabler of C-V2X use-cases with the application areas, including Ultra-Reliable Low Latency Communication (URLLC), enhanced Mobile Broadband (eMBB), and massive Machine Type Communication (mMTC). URLLC serves the low-latency and time-critical needs, while eMBB can handle the high-bandwidth scenarios, although traffic information may also require covering mMTC needs for high endpoint density. Existing standards and recommendations mainly reflect the bandwidth, latency, and jitter requirements with respect to generic use-case requirements [41], and especially the needs of automated vehicles [16]. The 5G architecture [42] was created with these requirements in focus.

Accurate and applicable metrics for autonomous vehicles need to be developed, while mobile network operators provide the required Quality of Service (QoS). The biggest concern is whether the 5G network solution suits the demands of the automatic use of vehicles. Another significant advantage is the smooth handover between cells without packet loss, in addition to the greater usable bandwidth that is provided by 5G in contrast to non-cellular technologies. 3GPP standards specify the key V2X scenarios and use-cases in the following areas with diverse performance requirements [16], including general use-cases, platooning, support of vehicle QoS estimation, as well as advanced and remote driving.

### 3.2. Expected Results and Challenges

In general, there are two pathways for the transition to next generation cellular networking: 5G Standalone (SA) and 5G Non-Standalone (NSA) architectures. In the case of SA, a new 5G Packet Core needs to be introduced with several new capabilities that are built inherently into it. The SA architecture comprises of 5G New Radio and 5G Core Network. In contrast, NSA allows for operators to leverage their existing network architecture mobile core instead of deploying a new core for 5G. Therefore, most of the Mobile Network Operators deployed some variants of 5G NSA, in the case of the current architecture and measurement campaign Option 3.X (Figure 21). The most critical aspect of this architecture regarding the measurement results is that only the downlink channel is 5G capable; the uplink still remains at 4G [43]. Therefore, the RTT (Round-Trip-Time) values will be higher than in the case of 5G SA with 5G Packet Core; however, our initial assumption was that this NSA deployment could already fulfill some of the requirements of [16].

### 3.3. Measurement Architecture

A non-commercial 5G modem—still under development phase—was used for the measurement, while the packets were generated on three end-devices. Three measurement traffic generators (device A1, A2, A3) were installed in a moving vehicle (in the M2 type BMW). The measurement traffic was generated on the end-devices (A1, A2, A3) and traveled towards the UE modem via Gigabit Ethernet connection. The modem was responsible for the 5G NSA Radio Access Network connections. From the UE modem, the packets always traveled through the (4G) eNB on the uplink channel towards server B. The next mobile network entity in the connection is the EPC (Evolved Packet Core); after that, the packets arrived at server B. Backwards on the downlink channel, the packets transmitted on a wired connection through the EPC to the (5G) gNB. Finally, via the UE modem, the packets returned to the end-devices on the Gigabit Ethernet interface.

### 3.4. Measuring the 5G Downlink NR Latency

End-to-end Round Trip Time (RTT) does not characterize the 5G capabilities of the architecture well, as only the downlink channel was 5G capable. Furthermore, one-way downlink latency merely offers a partial characterization of the network. The latency between the UE and the gNB is the most crucial parameter. However, we cannot measure this connection directly, as the protocol encapsulation makes it impossible. 3GPP based data-plane encapsulation is established between the UE and EPC. Neither the eNB nor the gNB are aware and allowed for opening the packet for encapsulation (in the case of close access to the eNB/gNB software, this could be monitored, but we did not have access). We only had access to the interfaces of EPC, UE, and the measurement servers and measured RTT, which is the sum of uplink and downlink latency. However, the gNB-EPC latency characterizes the network capabilities of 5G in contrast to the previous mobile networks. The supplementary connections are as follows in Figure 21:1-way latency ($T_{0}$): latency between (*l-bw*) A1 and B.Connection 1 ($T_{1}$): *l-bw* the UE’s LAN interface (i/f) and the end-devices.Connection 2 ($T_{2}$): *l-bw* UE’s WAN i/f and end-devices.Connection 3 ($T_{3}$): *l-bw* EPC SGi i/f and server B.Connection 4 ($T_{4}$): *l-bw* eNB’s s s1u i/f and server B.Connection 5 ($T_{5}$): *l-bw* gNB’s s1u i/f and server B.

### 3.5. Measurement Method

Our tests were carried out under different radio conditions. To define some of the primary parameters of the 5G network, we selected a state-space approach. One of these was the speed of the car controlled by cruise control (Figure 22 and Figure 23)—while the vehicle speed, the Inter-Arrival-Time (IAT) between the packets, and the Packet Length (PL) varied, latency was measured. Because most of the V2X use cases have strict time-critical specifications, these are among the most basic characteristics in potential 5G vehicle use-cases.

Based on our previous works in [44], we identified measurement scenarios with different PL and IAT parameters. Besides, one reference scenario was measured with constant parameters. The examined state-space scenarios are as follows:Scenario 1: from 2 ms IAT and 60 Byte PL to 62 ms IAT and 960 Byte PL, incrementing 60 Byte PL by every iteration and 20 ms IAT by every fourth iteration;Const: 2 ms IAT and 40 Byte PL.

The RTT results are presented as box and whisker plots. As expected, the outlier values are more widespread as the car speed increases (Figure 23). A possible explanation for the widespread whiskers is that, in the case of higher IAT, after a transmission, the radio bearer connection releases. Before the next packet, the radio bearer connection has to be set up again, which takes some milliseconds. However, this topic needs further examination. Figure 24 presents the RTT distribution for the mixture of different packet sizes. It does not indicate such a pattern as the previous one, but this measurement method can serve as a reference measurement.

The more detailed evaluation of the related 5G measurement results has been included in the meantime in another work of ours [13].

### 3.6. Future Works

Several network parameters need to be examined to evaluate 5G networks’ current capabilities regarding 5G-based self-driving vehicles and V2X in general, including functional properties and mass-behavior of these vehicles. The focal point of 5G V2X communication is the analysis of conditions under which the 5G network resources meet the service requirements. Particularly, if some QoS of the V2X use-case cannot be guaranteed by the 5G network (bandwidth, latency, packet loss, availability, etc.), the services that are based on them cannot be guaranteed either. However, 5G and mobile networks offer possibilities to improve the best-effort service structure with network slicing and Non-Public-Networks (NPN). Network slicing is a feature of 5G, which enables the multiplexing of virtualized and independent logical networks on the same physical network infrastructure. Moreover, slicing [45] promises to reach end-to-end service guarantees for the given services. While the Non-Public Network is a unique service of mobile network operators, it is to be implemented in a limited geographic area. The NPN infrastructure is implemented according to the use-case requirementsm and the service is provided based on the agreed SLAs (Service Level Agreements).

Currently, 5G network slicing is not yet available, while public mobile networks cannot offer any SLAs due to the best-effort operation. However, NPNs provide an excellent solution to test and develop V2X services close to real-life conditions before the commercial deployment of network slicing. These tests can be very beneficial, as the theory often does not reveal environmental, hardware, and software synergies that are immediately emerging in real-life tests and verification processes.

Our future work can focus on testing V2X services in the 5G NPN environment. The main parameters of the examination can be the user density and dynamic environment aspects (distance change of the radio tower, vehicle handover, and weather conditions).

## 4. Measurement Results with Application Examples of the LIDAR, Camera, Radar, and GNSS Devices

At the beginning of the measurement campaign, initial position data records with all of the participants GNSS systems were taken, in order to eliminate failure possibilities. Additionally, the aim of these measurements was to determine where is the acquired position (which was given by the GNSS system) in relation to the vehicle itself. For this measurement, a preliminary defined marker point was used. The accurate position of the marker point was taken before the measurement campaign. Each participant car was ordered to park to the marker point with the same fashion, positioning the center of its license plate exactly above the marker. From these data, a chart was formed, containing the position offset information from the center of the front license plate for every participating vehicle.

During the three days measurement campaign held at M86 highway, there were 41 executed scenarios, containing 137 test runs. The scenarios were planned to satisfy the special requirements of each participating partner. These traffic situations can be ordered into five groups, which are the following: scenarios that support radar-based measurements, such as calibration and object recognition. Scenarios for camera image-based object detection and traffic situation evaluation during day and night-time. Scenarios for LiDAR and camera sensor fusion-based object detection and traffic situation evaluation. Scenarios that support object detection with stationary mounted infrastructure sensors and provide the opportunity for distributed data recording system tests. High speeding or dangerous scenarios require a controlled environment. These scenarios were carried out in a way that they were mutually beneficial for multiple test purposes. For example, the infrastructure sensors could also make valuable recordings during scenarios for radar measurements. The size of raw measurement data, including LIDAR point clouds, camera pictures, and radar information exceeded 1500 GB.

### 4.1. GPS Data Features

All of the participants used RTK correction service to push down the position inaccuracy below 5 cm or less, based on the employed GNSS system. The GPS data acquisition frequency was defined to 100 Hz for every participant, which gives a good position resolution through time even in large speed scenarios. The exported GPS logs contained information regarding time, position, heading, and quality. The quality information comprised standard deviation values or solution status and number of satellites in view. At the post processing work, the standardization of the collected position log data was the first step. This was necessary because of the great number of different GNSS devices and data capturing solutions, using large variety of time, heading, and position formats. After standardization, the logs were evaluated by the quality information which they held. A graphical representation was made in order to quick check whether there were any unexpected quality drops during a test run. Subsequently, a summary was made, showing quality information of all the vehicles, participated in a given test run. Figure 25 holds an example summary of quality information form a test run. The quality information was shown in the same time scale for every participant, enabling comparison of the logs with each other.

Next, the quality log, which had the most impact for the given test run, was selected and plotted at a finer time resolution. With this information, time intervals of good or poor quality could be declared. Finally, the position information was exported to Google Earth Pro supported format, with 1 Hz sample rate. Joanneum Research supplied a Google Earth Pro compatible HD Map version of the M86 highway section. With the map and position information, the trajectory of each participant and, therefore, the whole test run, could be visualized and replayed through time. The visualization results for a given test run are presented in Figure 26.

Two deployed infrastructure station was used, each built up from a LIDAR sensor, and two cameras facing parallel with incoming and outgoing traffic directions. Figure 27 shows a point cloud and camera image from the same moment, captured by infrastructure sensors.

The data acquisition system used as infrastructures was time synchronized with each other with millisecond accuracy. One of the participant cars, namely the Honda CRZ from Budapest University of Technology and Economics, was also receiving the mentioned synchronizing signal alongside the infrastructure systems. This feature enables seeing an exact moment in time with two, sometimes even three different points of view, when the Honda participated in the given test run as Figure 28 shows.

### 4.2. Ground Truth Information for Object Detection Algorithms

The shared GPS data was captured with high accuracy (Figure 29 presents a ground truth validation to a captured camera image), 100 Hz resolution, and has detailed quality information. The participants took their own measurements such as LIDAR, radar or camera recordings in addition to the shared GPS data. They can use their respective data as they see fit. The Budapest University of Technology and Economics (BME) made recordings with the sensors deployed for the infrastructure, and on the Honda vehicle as well. The acquired point clouds and image recordings, combined with the shared position information, generate a data set with ground truth information labels. This data set can be used for testing and validating neural network based object detection algorithms.

For contribution to global scientific research, the Budapest University of Technology and Economics published a part of this data set to aid the development of neural network based object detection projects or any other automotive testing. The data set can be found at https://automateddrive.bme.hu/downloads (accessed on 19 March 2021).

### 4.3. Example Application—Determine Object Position Based on Homography

An example detector application is presented to point out a way of profiting from the ground truth data and the created UHD map from the motorway section. In this case, the processed data came from one of the infrastructure mounted cameras. The footage was a series of timestamped image frames. A YOLO4 [46] detector processed each image and gave bounding boxes for each vehicle that it could detect. The detector also gave the pixel coordinates of the middle point of the bounding boxes’ lower border. Figure 30 presents a result of object detection. The pixel coordinates were combined with the corresponding timestamps of the pictures.

A conversion transformation had to be defined in order to produce GPS position information from the pixel coordinates. The road surface was considered as a plane, therefore projection transformation from the image plane to the road surface could be calculated. The method called homography and the transformation was described by the homography matrix. To obtain a good estimation of the matrix, coordinates of at least ten points (shown in Figure 31) must be known in both planes. Distinctive points were selected in an object free image frame. The pixel coordinates were manually collected with a basic image editor program. The corresponding GPS coordinates could be determined by using one of the UHD maps. The map could be visualized, as shown in Figure 5; therefore, the previously selected distinctive points could be found in the map as well. After the point coordinates were measured in both planes, the homography matrix was estimated.

Using the homography matrix, the detection results were transformed into GPS coordinates. Based on the timestamps, the detection was assigned with ground truth information. Each detection was paired with the ground truth, where the time difference between them was the least. This resulted in a maximum of 5 ms time shift between detection positions and corresponding ground truth information. The offset between the position given by the vehicles’ GNSS system and the position of the front license plates middle point was considered. Figure 32 shows the described process.

Figure 33 shows the distance between the detected object position and the position from the corresponding ground truth.

In the camera point of view, when the detected object was afar, the front of the vehicle was close to the GNSS reference point, so the two positions were close together. When the vehicle came closer, the distance of the front and the GNSS reference point became greater, being closer to the real value. Adding the offset caused a high position difference at the beginning of the footage, but, as the vehicle approached the infrastructure, the position deviation lessened. Figure 34 shows this phenomenon, where the position information provided by the YOLO detector and the ground truth were presented in relative to the sensor itself.

Error metrics were applied to the detection results. First, the root mean square error was determined according to the formula shown below
(1)$$RMSE=\sqrt{\frac{1}{n}\sum_{i=1}^{n}{\left(x_{i}-{\hat{x}}_{i}\right)}^{2}},$$
where $x_{i}$ stands for ground truth and ${\hat{x}}_{i}$ represents the position determined with YOLO detector and homography transformation in each frame. The calculated error value was 1.499 m for the whole recording. Subsequently, the mean absolute percentage error was computed. In the equation below
(2)$$MAPE=\frac{1}{n}\sum_{i=1}^{n}\left|\frac{\left(x_{i}-{\hat{x}}_{i}\right)}{x_{i}}\right|,$$

$x_{i}$ and ${\hat{x}}_{i}$ represent the ground truth and YOLO detection distances relative to the camera, respectively. For the whole recording, the mean distance deviation between the measured and ground truth positions were five percent of the objects distance from the camera. The location of the camera was obtained from one of the UHD maps of the measurement site.

The presented example application shows the value of the collected ground truth information, and the high definition digital maps. With the recorded data that were collected from numerous different sensors, more sophisticated tools can be developed, producing extensive and more accurate information from the detected objects.

### 4.4. Example Application—Determine Object Position Based on Camera LIDAR Fusion

Several algorithms are available to detect objects, both on a camera and LIDAR basis. Both of the sensors have their pros and cons. Camera-based systems are extremely fast, thanks to modern graphical hardware. Several traditional and machine learning algorithms are also available. For the time being, we can find less mature ones from the algorithms that process LIDAR data. The advantages of camera-based systems include fast and accurate detection of objects. The advantage of LIDAR-based systems is the determination of accurate spatial coordinates, even in low light conditions. In our presented detector, we combined the advantages of the two systems. In this example, the position coordinates of objects are yielded by a low level sensor fusion approach. This technique uses the data from a camera and a LIDAR sensor, both mounted on a measurement vehicle used in the tests during the measurement campaign.

The detector consists of two parts. In the first part, the camera image is processed by the YOLOv4 algorithm, which defines bounding boxes for the detected objects. The system recognizes all objects taught in the COCO data set, including those that are necessary to us: pedestrians, cars, trucks, buses, etc. The second part of the detector processes the output data of the YOLO block. This block uses detections, an image, and a point cloud as input. On vehicles, it also uses position data from the GPS. The accurate calibration of the camera and LIDAR is critical to the detector.

The detector first projects the point cloud onto the camera image (as shown in Figure 35), from which it determines the pixels in each bounding box and the corresponding 3D points. We associate these points with the objects. However, with this method, many outlier points can be placed in the boxes (especially for pedestrian detection). To eliminate these, we implemented several solutions in the algorithm. One solution uses statistical methods to filter out points that do not belong to the object. The other method uses a small window inside the bounding box. The size and location of the window depend on the type of object detected. It then determines the point in the window—the farthest for vehicles and the closest for pedestrians—and leaves points outside a set distance range. This quickly filters out false object points. In the last step, we determine the geometric center of the filtered point clouds, which is transmitted by the detector as a message with a defined structure. An object tracker with Interactive Multiple Model filtering was used to create the logical connection between detections in each frame through time. The resulst of the tracking is presented in Figure 36. The tracker also filtered out false positive detections given by the YOLO algorithm.

In this recording, the Green Truck from Knorr-Bremse and the M2 type BMW from BME were followed and position information was determined for these two objects. The tracker gave the object positions in a coordinate system, where the IMU (the Inertial Measurement Unit of the on-board GNSS system) is the origin of the measurement vehicle.

The positions were transformed into a global coordinate system with the ground truth information regarding the measurement vehicle itself. During the process, the heading information of the ego vehicle was considered in order to determine the correct position of the detected objects in the global coordinate system.

After the transformation process, the detection and ground truth data were paired with each other, based on the time stamp data that they had beside the position information. Subsequently, the distances from the ego vehicle to the detected and to the ground truth positions of the target vehicle were also calculated in case of the BMW and the Truck. Figure 37 shows the process of data analysis and Figure 38 presents the comparison results.

The time resolution for the ground truth data is 10 ms. The recorded data has a different resolution, therefore some inconsistency arose during the pairing. This inaccuracy is no more than 5 ms. During the processed time interval, the target objects are accelerating from a near-standing state to a moderate speed. When the speed of the vehicles increases as shown in Figure 39 this time shift has a growing effect for the results.

The difference between the detected position-ego vehicle and the ground truth position-ego vehicle distances represents the accuracy of the application example. The mentioned difference values can be seen in Figure 40. The effect of the time shift was recognizable on the results. Therefore, in case of the BMW the vehicle speed was greater, the time shift effect was higher than in the case of the Truck.

The error metrics were calculated for the results. For the BMW, the Root Mean Square Error was less than the Truck’s case, but this trend flipped when the error was considered in relation with the distance, as the Mean Absolute Percentage Error values pointed out. Table 1 shows the error metric values.

## 5. Conclusions

In midsummer of 2020, a grandiose measurement campaign was fulfilled on a real-world motorway stretch of Hungary with the effective collaboration of international industrial and academic partners. Detailed data collection was carried out using both vehicle and infrastructure sensors. The obtained results will be especially useful and applicable for future automotive R&D activities, due to the high-precision of the logged data.

The main lessons that were learned and experiences of the measurement campaign are as follows:Planning and managing a measurement campaign with several partners and with different sensors is a huge and complex task where success relies on thorough preparation.For high precision mapping, two datasets were collected using different high-tech instruments during the measurement campaign. Different capability of sensors are needed when identifying small details, such as traffic supplemental signs or when creating a surface model of the ground.Ground truth information for object detection algorithm is of crucial importance in the automotive testing field. The acquired point clouds and image recordings combined with the shared ground truth position information can be directly used for testing and validating neural network based object detection algorithms.The presented two application examples demonstrate the viability of the collected data during the M86 Measurement Campaign. This data set may support a large variety of solutions, for the test and validation of different kinds of approaches and techniques.5G tests were carried out under different radio conditions. Different measurement scenarios provided latency results that behaved as expected beforehand.

As a continuation of the presented measurement campaign, further application examples and research results will be published by the participated partners by leveraging the measured data.

## Figures and Tables

**Figure 1 sensors-21-02169-f001:**
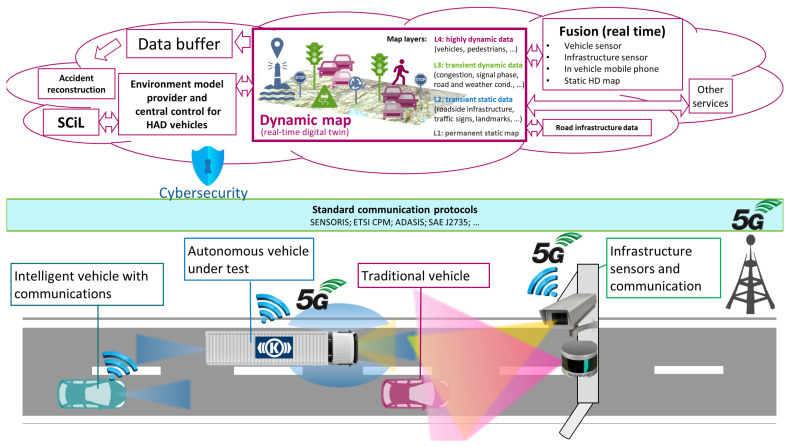
The goals of the international project “Development of central system architecture for autonomous vehicle testing and operation related services (2019-2.1.14-ÖNVEZETŐ-2019-00001)”.

**Figure 2 sensors-21-02169-f002:**
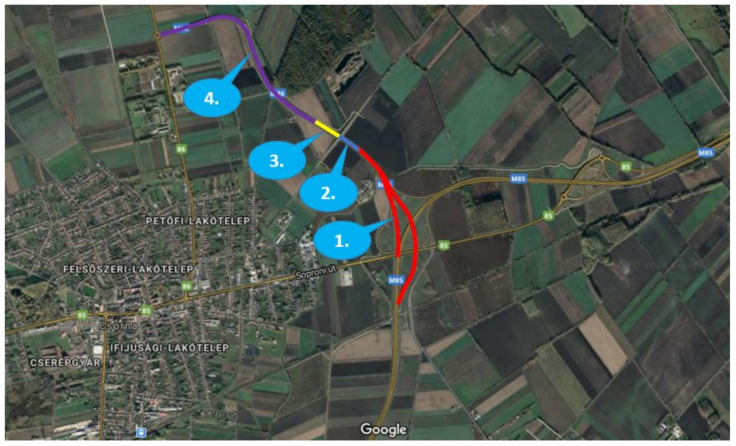
Sections of the test site (3.5 km in all) located near Csorna city (Hungary) on route E65 (GNSS coordinates: 47.625778, 17.270162)

**Figure 3 sensors-21-02169-f003:**
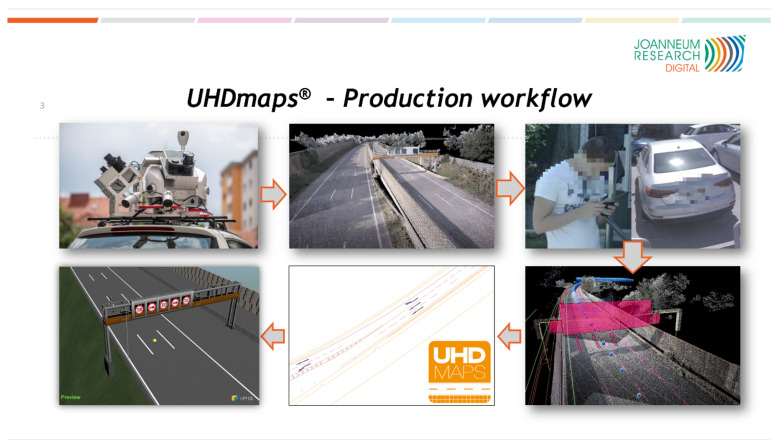
Digital Twin production workflow which is an ongoing subject of research at Joanneum Research.

**Figure 4 sensors-21-02169-f004:**
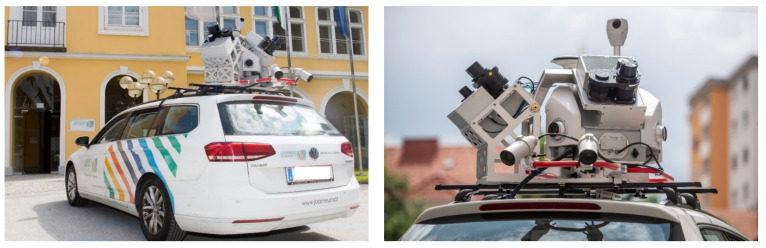
Leica Pegasus Two Ultimate with dual scanner configuration and two pavement cameras (Picture: © Bergmann, 2018).

**Figure 5 sensors-21-02169-f005:**
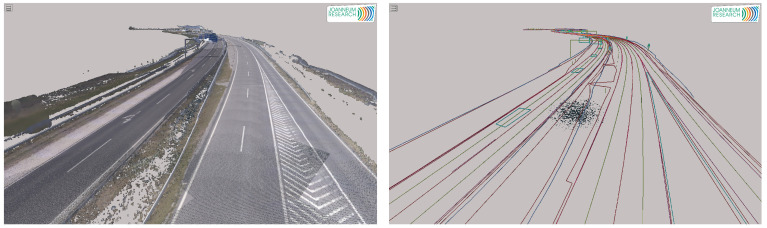
Mobile Mapping Survey data (**left**). Deduced UHDmap (**right**).

**Figure 6 sensors-21-02169-f006:**
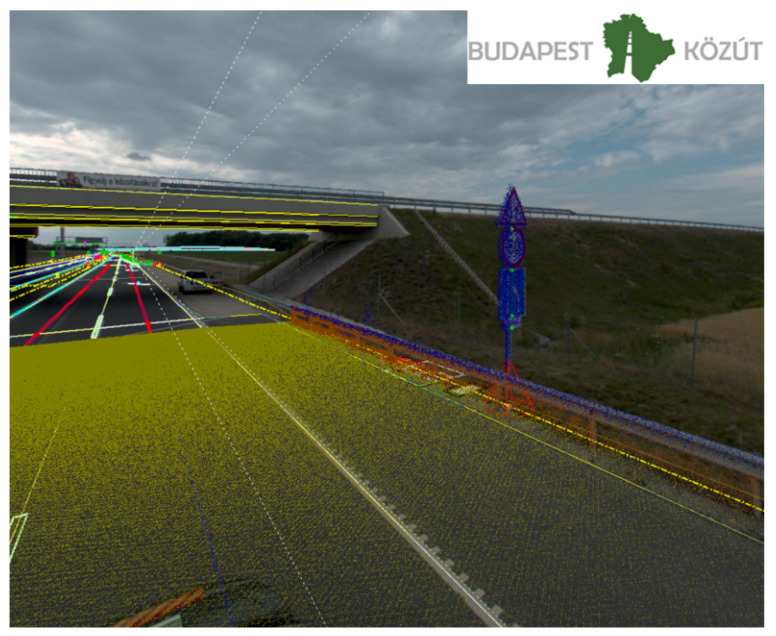
Three-dimensional (3D) point clouds and camera picture with 3D vectors.

**Figure 7 sensors-21-02169-f007:**
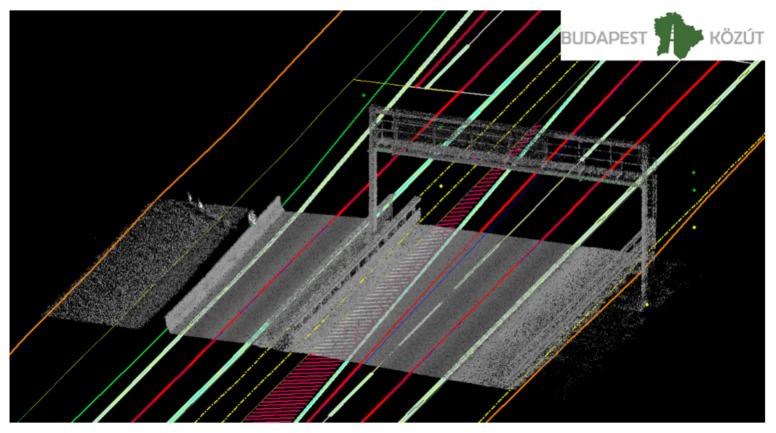
3D vectors for HD map.

**Figure 8 sensors-21-02169-f008:**
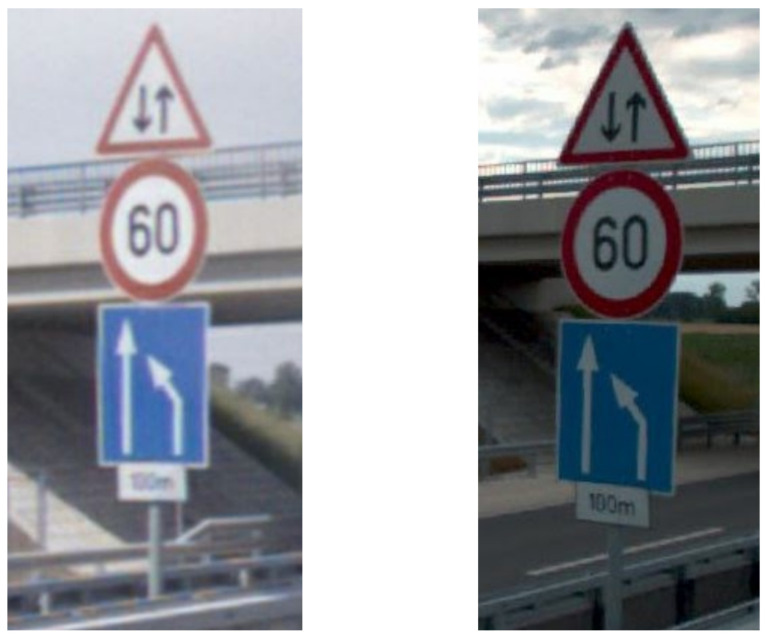
Shooting from the same spot (**left**): Leica Pegasus 2 Ultimate 360${}^{\circ }$ panoramic photo crop, (**right**): Ladybug 5+ (Riegl) Camera Picture crop.

**Figure 9 sensors-21-02169-f009:**
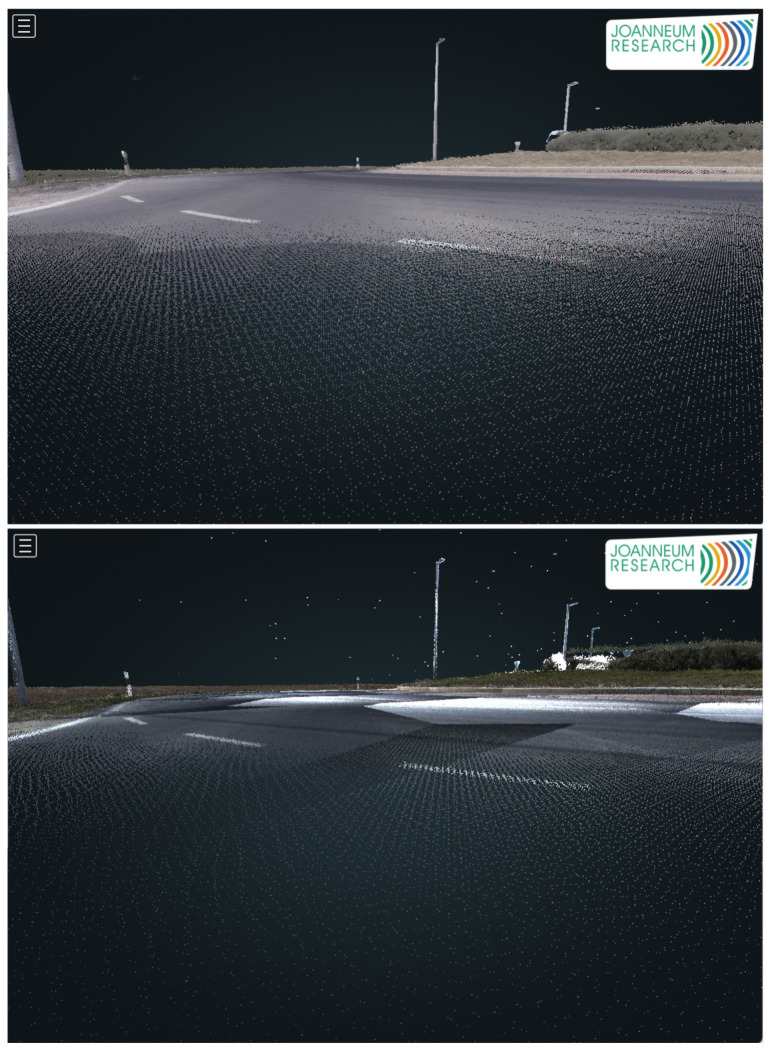
Comparison between the Z+F 9012 (**top**) and the Riegl VQ-450 (**bottom**) while scanning ground level of an exit arm in a roundabout.The reduced point density of the Riegl compared to the Z+F scanner is clearly visible. Both systems are dual profiler setups.

**Figure 10 sensors-21-02169-f010:**
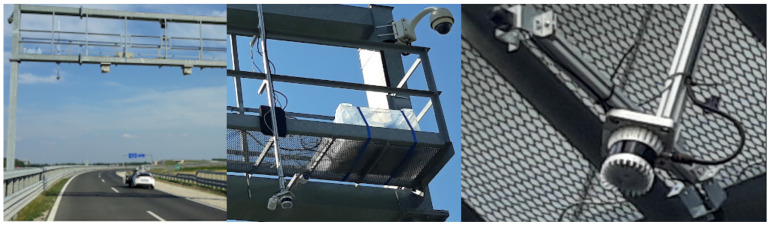
Infrastructure setup. Left to right: Infra1 from a distance, Infra1 from the other side, and Infra2 sensors close-up.

**Figure 11 sensors-21-02169-f011:**
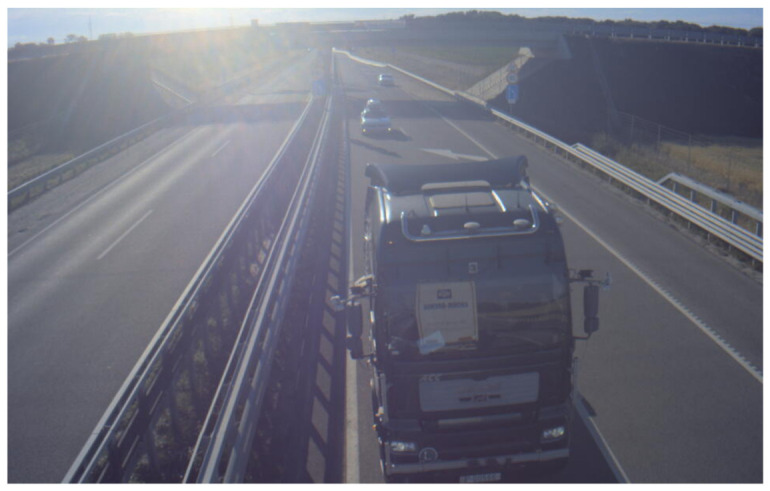
Example image taken from Infra2 station, facing the overpass.

**Figure 12 sensors-21-02169-f012:**
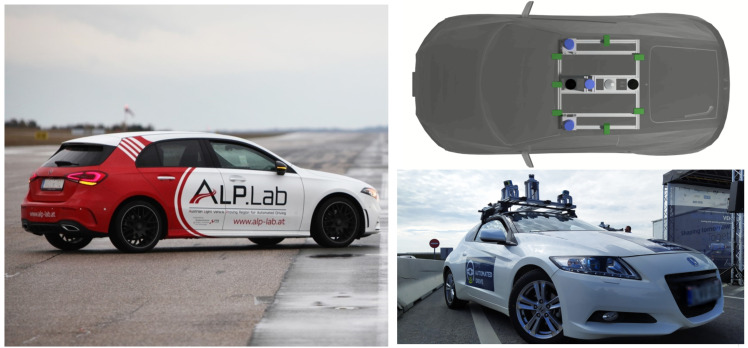
ALP.Lab test vehicle used during the measurement campaign (**left**) and sensor setup of the Budapest University of Technology and Economics (BME) measurement vehicle (**right**).

**Figure 13 sensors-21-02169-f013:**
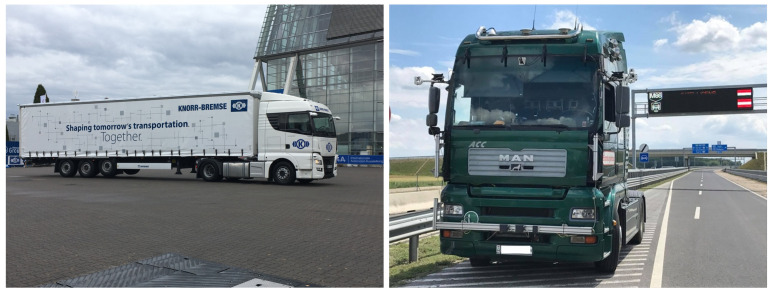
Knorr–Bremse tractor–semitrailer combination (**left**). Solo tractor test vehicle (**right**).

**Figure 14 sensors-21-02169-f014:**
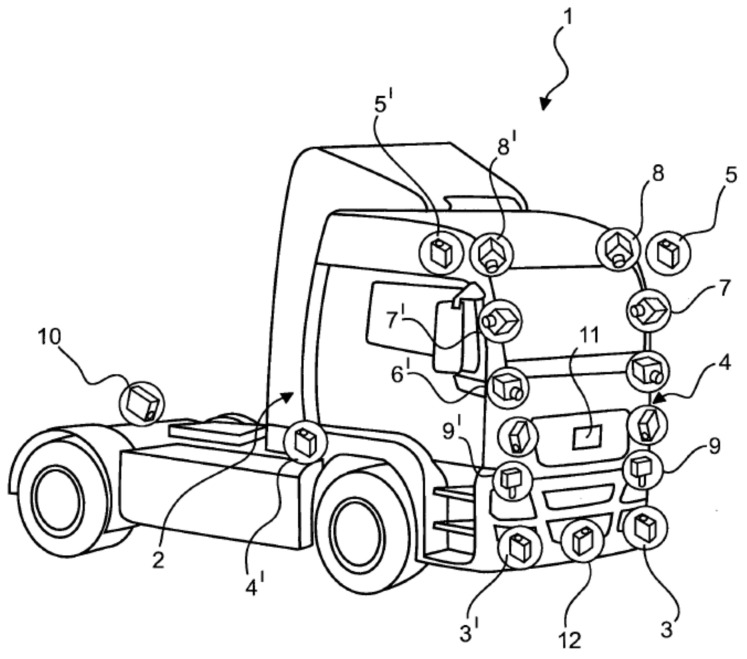
Knorr–Bremse environment sensor setup.

**Figure 15 sensors-21-02169-f015:**
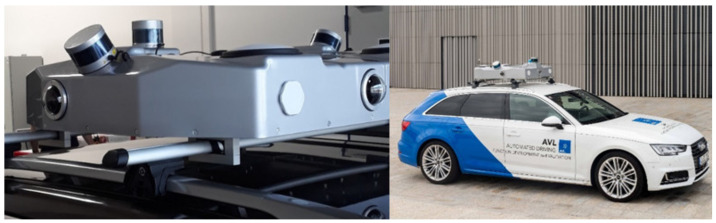
AVL Dynamic Ground Truth rooftop box.

**Figure 16 sensors-21-02169-f016:**
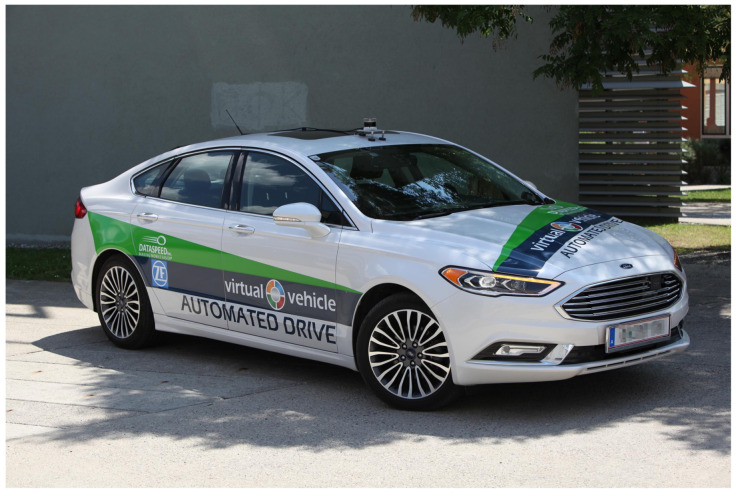
Virtual Vehicle Automated Drive Demonstrator (ADD).

**Figure 17 sensors-21-02169-f017:**
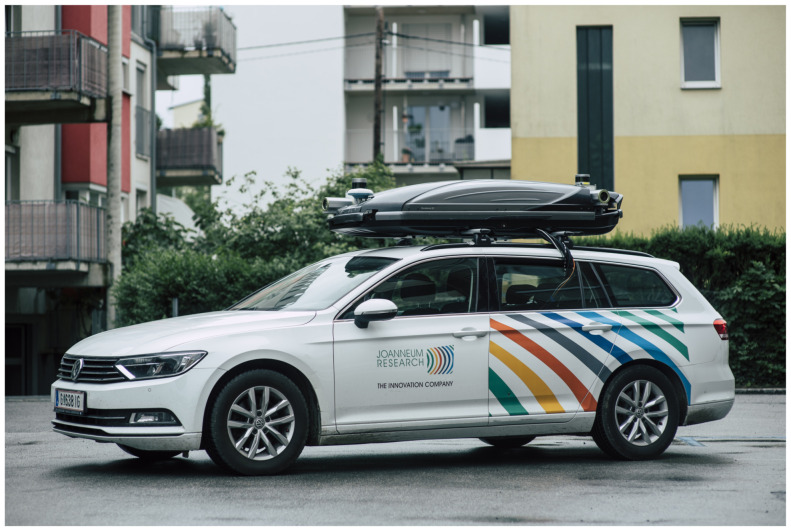
Joanneum Research DGT System. Photo ©Bergmann.

**Figure 18 sensors-21-02169-f018:**
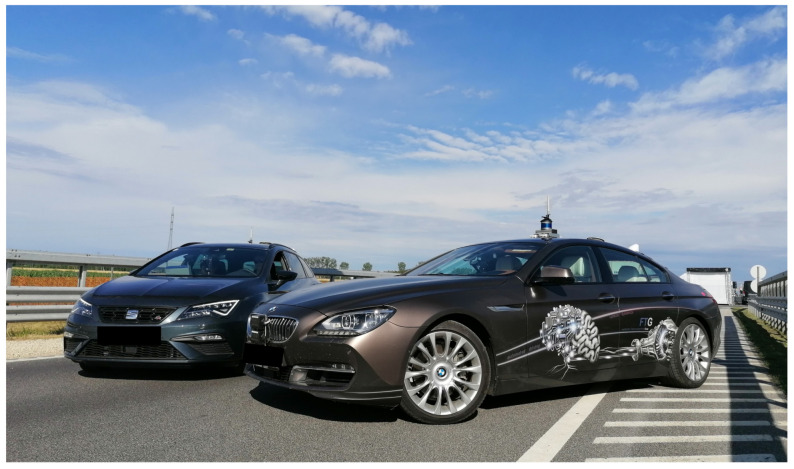
TU Graz vehicles: BMW 640i and Seat Leon.

**Figure 19 sensors-21-02169-f019:**
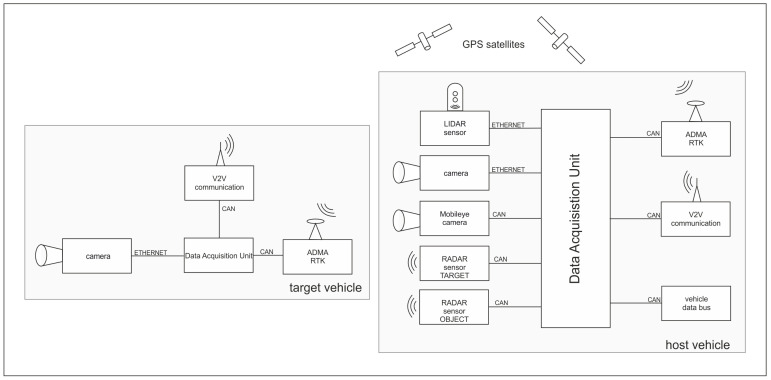
TU Graz measurement setup.

**Figure 20 sensors-21-02169-f020:**
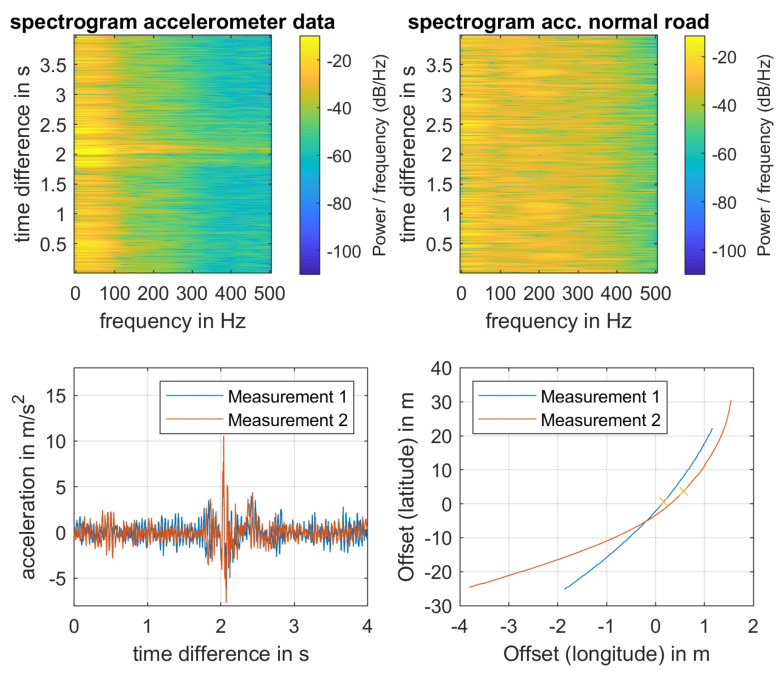
(**Top**): Spectrogram analysis of *z*-component of accelerometer data (Left: test-route, right: normal street). (**Bottom**): comparison of time-signals of *z*-accelerometer data and corresponding relative GPS position. The crosses in the GPS position plot indicate the relative time-stamp of 2s corresponding to the peaks being observed in the accelerometer data.

**Figure 21 sensors-21-02169-f021:**
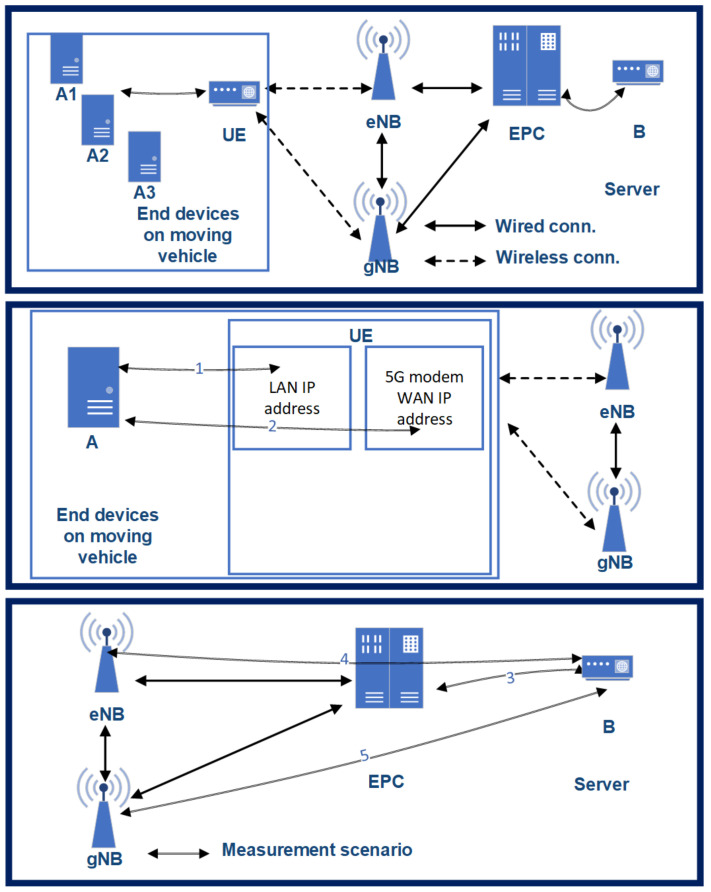
The architecture of the 5G testbed—as 3GPP Option 3X.

**Figure 22 sensors-21-02169-f022:**
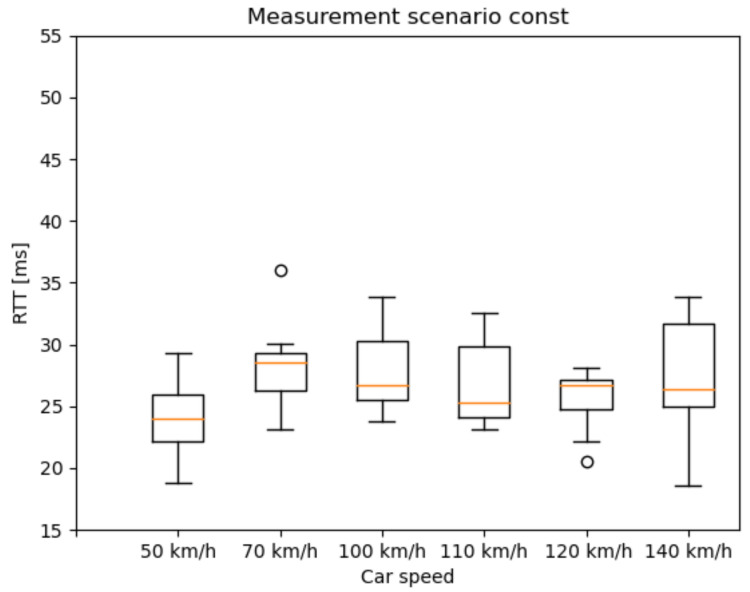
Latency results of the constant packet length and IAT measurement scenario.

**Figure 23 sensors-21-02169-f023:**
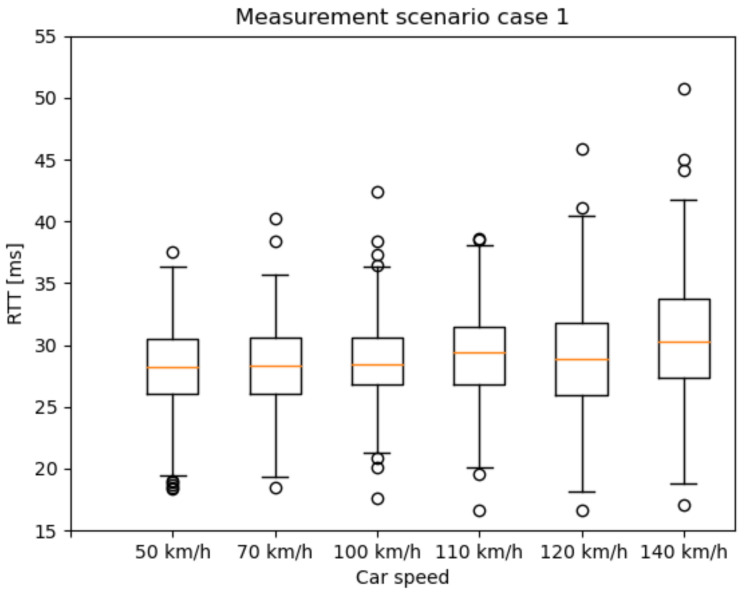
Latency results of the measurement scenario 1 with different vehicle speeds.

**Figure 24 sensors-21-02169-f024:**
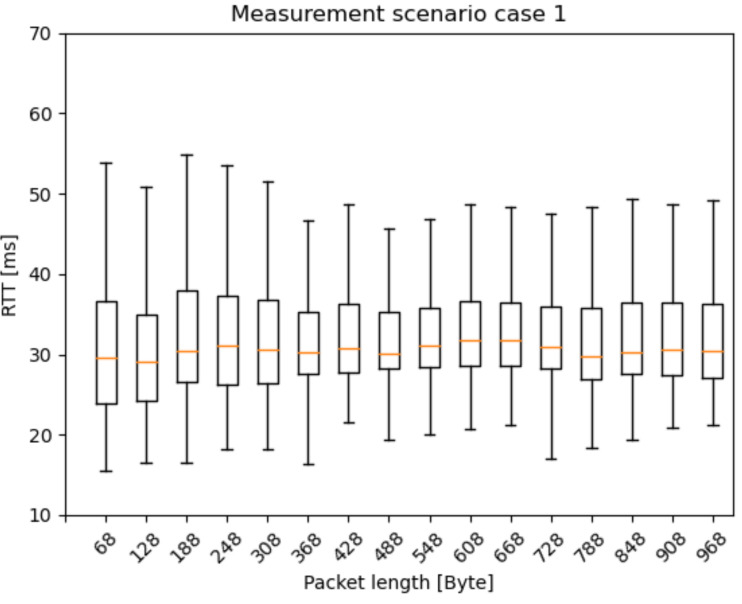
Latency results of the measurement scenario 1 with different packet lengths.

**Figure 25 sensors-21-02169-f025:**
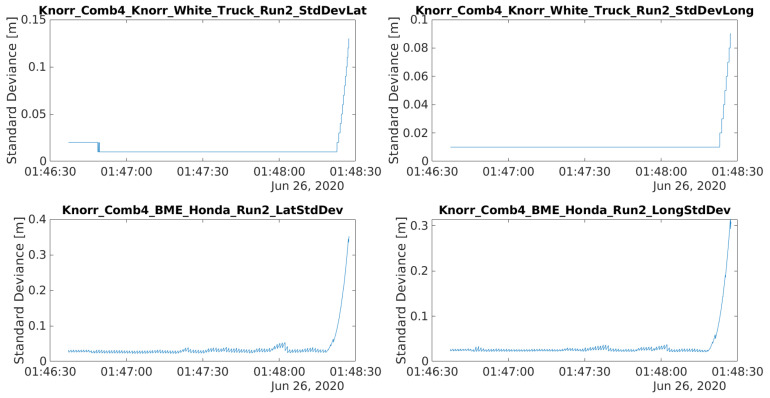
Quality information from all the logs for a test run.

**Figure 26 sensors-21-02169-f026:**
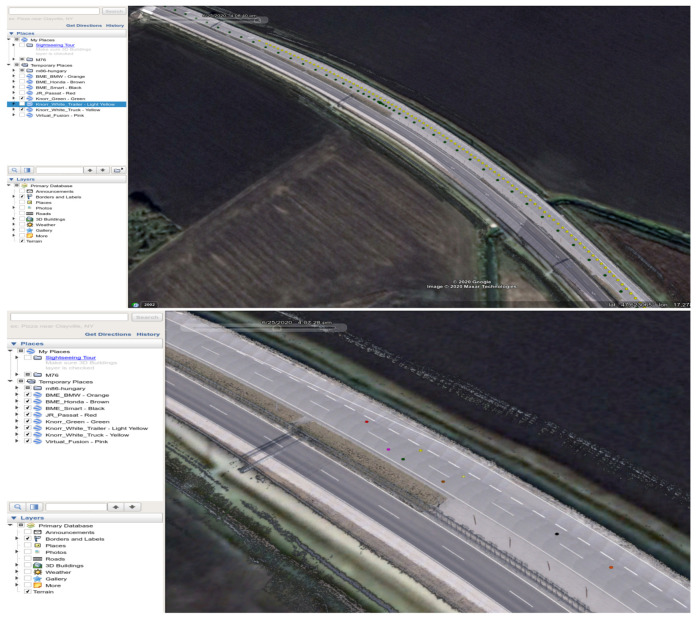
Plotted trajectories for the whole time interval of the test run (**top**) and for a given moment (**down**).

**Figure 27 sensors-21-02169-f027:**
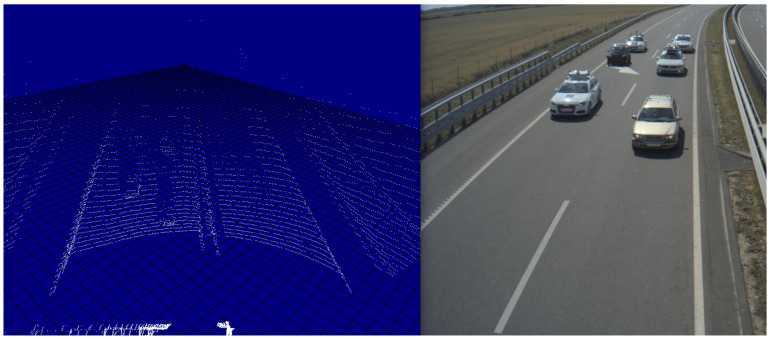
LIDAR point cloud (**left**) and camera image (**right**) from the same moment of time.

**Figure 28 sensors-21-02169-f028:**
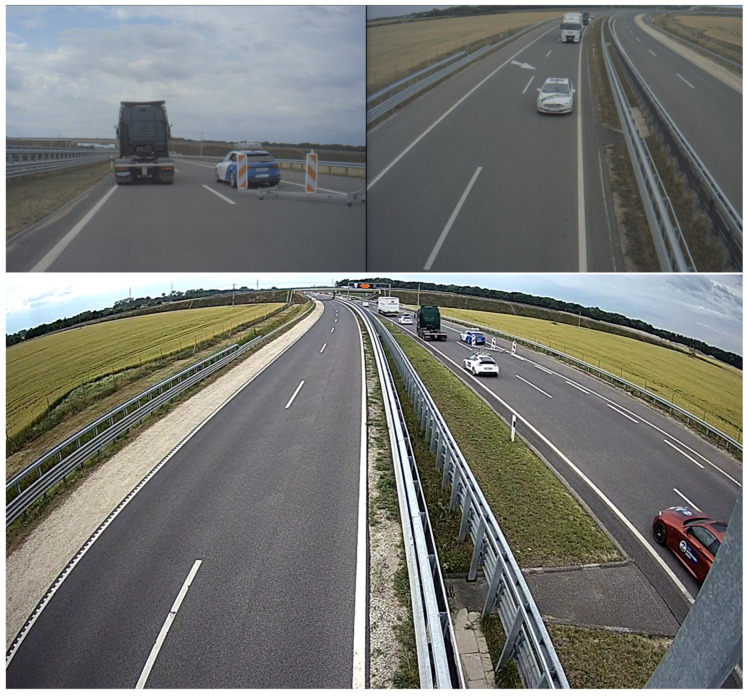
Pictures from the same moment of time from the point of view of the measurement vehicle (**upper left**) and the two infrastructure cameras (**upper right** and **down**).

**Figure 29 sensors-21-02169-f029:**
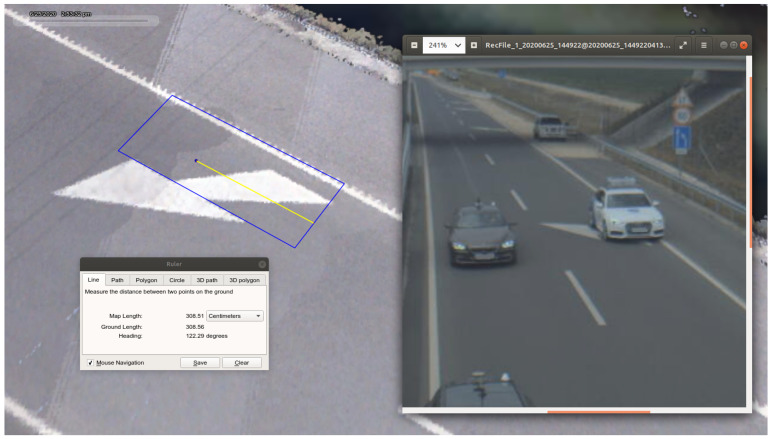
Outlines of white-blue Audi based on position data, plotted on UHD map of M86 section. The image was taken at the same moment. The position of the Audi on the image corresponds with the plotted outlines.

**Figure 30 sensors-21-02169-f030:**
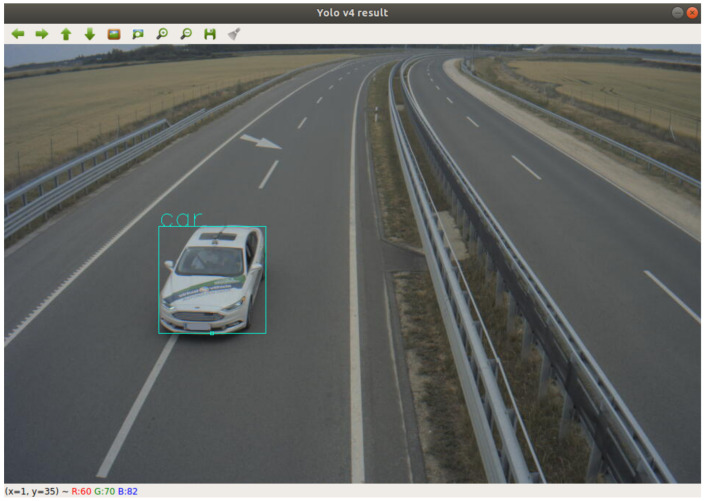
Bounding box drew around the detected object.

**Figure 31 sensors-21-02169-f031:**
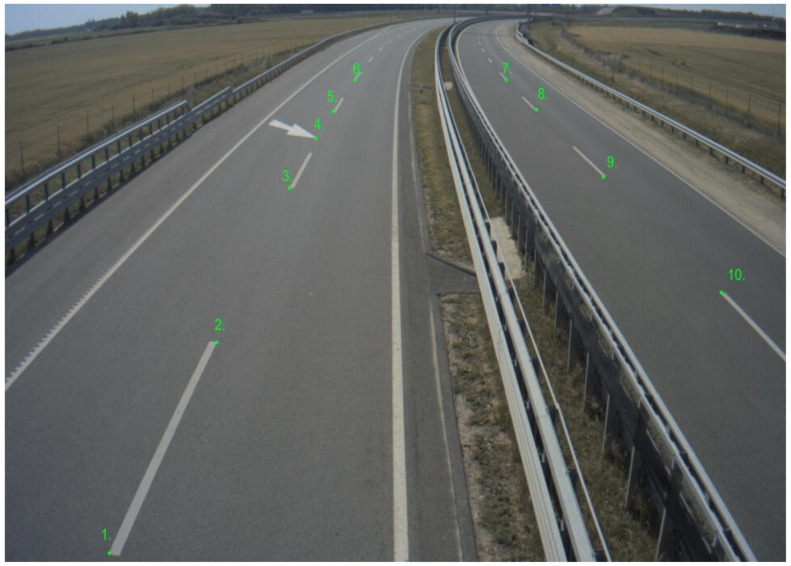
Points used to calculate homography matrix.

**Figure 32 sensors-21-02169-f032:**
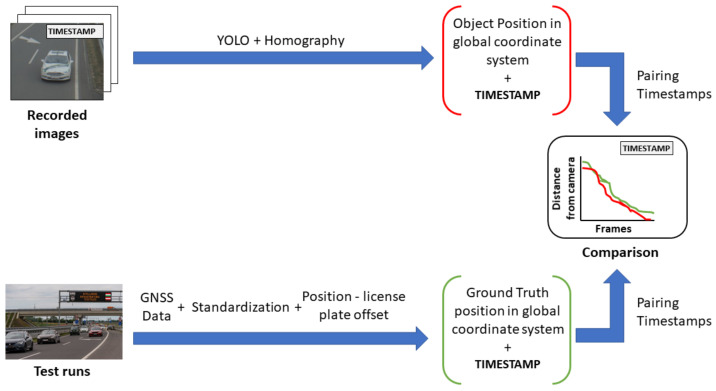
Evaluation process of homography-based object distance from infrastructure camera with ground truth data.

**Figure 33 sensors-21-02169-f033:**
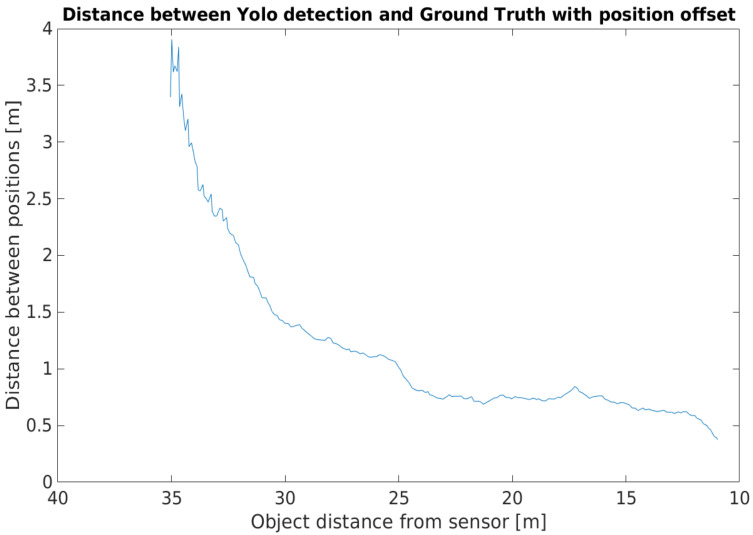
Position difference between ground truth and detection.

**Figure 34 sensors-21-02169-f034:**
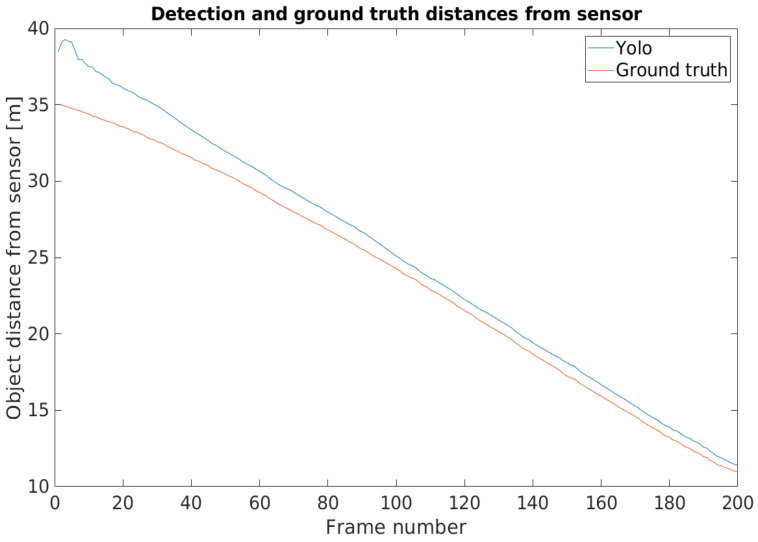
Positions relative to the sensor.

**Figure 35 sensors-21-02169-f035:**
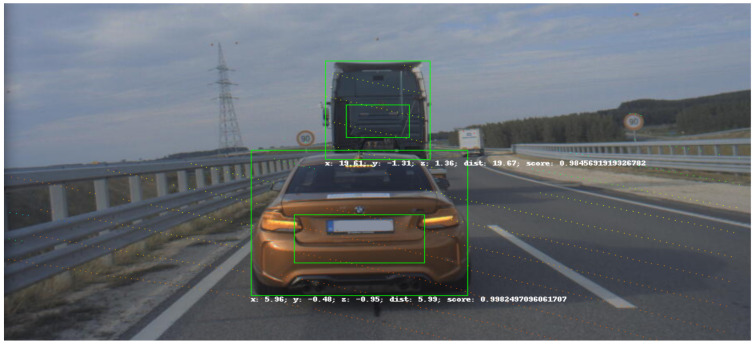
Point cloud projected into the image.

**Figure 36 sensors-21-02169-f036:**
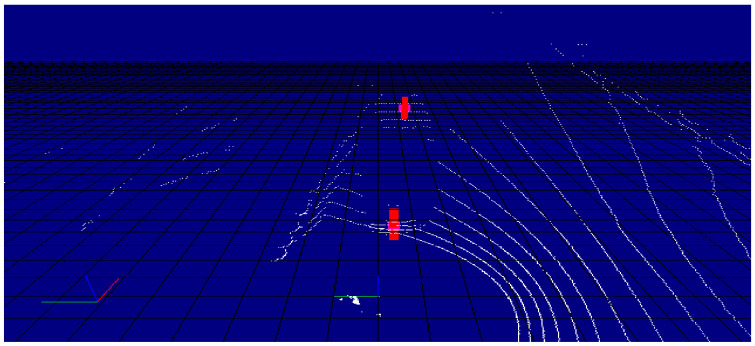
The tracked position represented with a red column in the point cloud.

**Figure 37 sensors-21-02169-f037:**
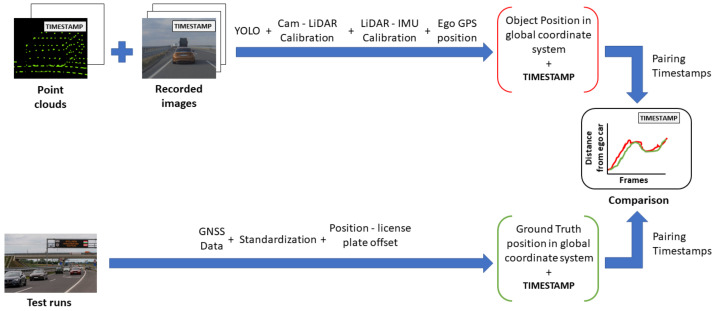
The evaluation process of fusion-based object distance from ego vehicle with ground truth data.

**Figure 38 sensors-21-02169-f038:**
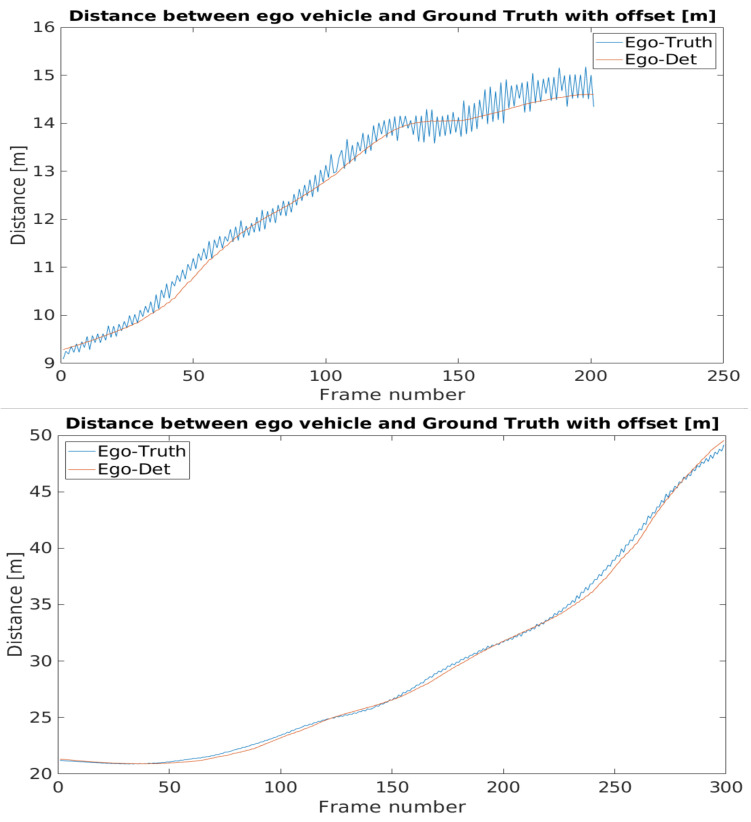
Distance of the detected and ground truth positions of the BMW (**above**) and the Truck (**below**) from the ego vehicle.

**Figure 39 sensors-21-02169-f039:**
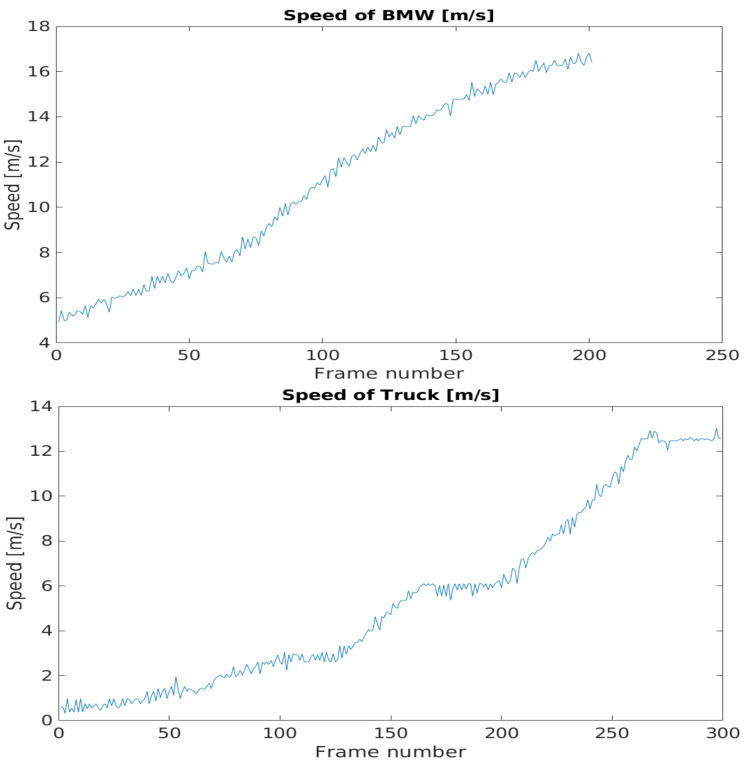
Speed values of the target vehicles: the BMW (**above**) and the Truck (**below**).

**Figure 40 sensors-21-02169-f040:**
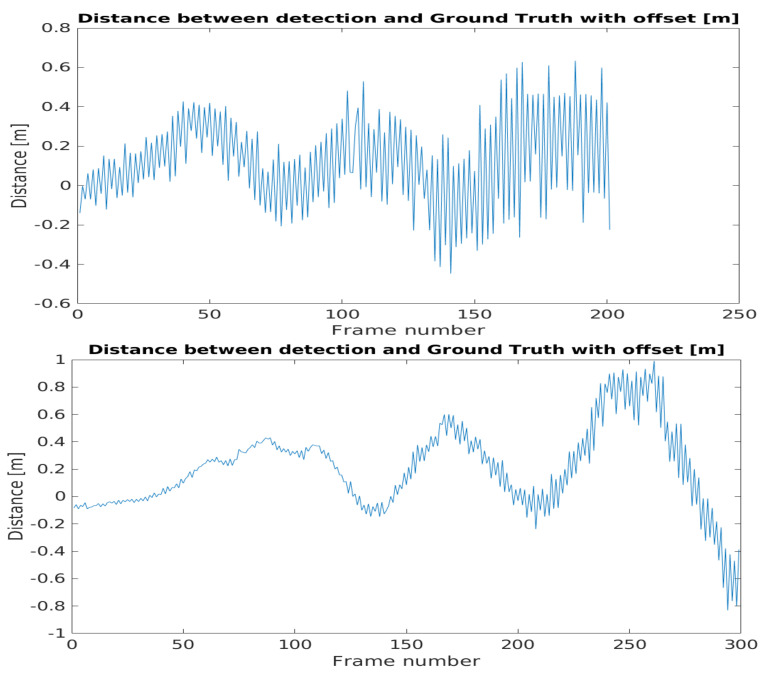
Distance differences in case of the BMW (**above**) and the Truck (**below**).

**Table 1 sensors-21-02169-t001:** Error metrics for the example application.

	BMW	Truck
RMSE	0.2631	0.3643
MAPE	0.0094	0.0070

## Data Availability

For contribution to global scientific research, Budapest University of Technology and Economics published a part of this data set to aid the development of neural network based object detection projects or any other automotive testing. The data set can be found at https://automateddrive.bme.hu/downloads.

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
