# Peer review of "Motorway Measurement Campaign to Support R&D Activities in the Field of Automated Driving Technologies"

_sensors, 2021, doi:10.3390/s21062169_

Round 1

Reviewer 1 Report

The paper presents the measurement campaign carried out on a real-world motorway stretch of Hungary. The structure of the paper is good. But to be publishable it needs major revision in presentation and to be more clear and usable for the researchers in the field, there are some issues that should be resolved to make the article publishable, otherwise, the work should be improved and resubmitted.

To improve the quality of the article, the authors should follow the given comments:    - The abstract should be improved. It should be more detailed on contributions of the presented work.  
  - There are lots of English typos and mistakes as on Page 2 line 26 'together' is repeated twice. Read the text well and correct all the typos.    - In Fig. 1 the block of the Dynamic map is not clear! it should be shown in a sub-image to help the details to be seen clearly.    - instead of showing just an image of how  Leica Pegasus is mounted on a car, a section should be added giving its details.    - The UHD map details should be clarified that what exactly a UHD map is, what lines are detected and how they are depicted. Detail the Fig 5-right  that what each line indicates?   - In the case of images shooting from the same spot but different cameras as Leica Pegasus 2 and  Ladybug 5+, the problem of image quality and noise can be resolved by applying advanced filtering techniques. This matter should be noted and values research works to be cited as "Mediated morphological filters"(2001)   - What exactly Figure 9 indicates? make it clear in detail.    - Figures 12, and 13 should be merged.    - In the introduction, one part should be added to discuss the role of IoT sensors in this technology, as IoT has a key role in the monitoring of vehicles as in the following research works its role has been given for the case of agriculture vehicles: "Economic data analytic AI technique on IoT edge devices for health monitoring of agriculture machines", "Economic IoT strategy: the future technology for health monitoring and diagnostic of agriculture vehicles".  

Author Response

Thank you for the thorough review and the valuable comments. 

We addressed all requested points and the corresponding corrections/extensions are emphasized by red color in the revised manuscript. 

1. The abstract should be improved. It should be more detailed on contributions of the presented work.  

[Reply 1.] 

The abstract was revised in order to better highlight the contributions of the presented work. 

2. There are lots of English typos and mistakes as on Page 2 line 26 'together' is repeated twice. Read the text well and correct all the typos. 

[Reply 2.]  

Thanks for your correction. The paper has been thoroughly revised and the typos have been removed. 

3. In Fig. 1 the block of the Dynamic map is not clear! it should be shown in a sub-image to help the details to be seen clearly.    

[Reply 3.]  

Figure 1 was redrawn with a much larger Dynamic map and larger fonts. All details now appear clearly visible at 100% zoom. Considering this, we believe the reviewer would not require a sub-image anymore, which was therefore omitted (but will be added if our assumption is incorrect). 

4. The UHD map details should be clarified that what exactly a UHD map is, what lines are detected and how they are depicted. Detail the Fig 5-right  that what each line indicates? 

[Reply 4.]  

A clarification on what is meant with UHD maps, in comparison to HD-maps, has been added to section 2.2.1. 

A clarification of the depicted map features (lines visible in FIG 5-right), has been added to section 2.2.1 (paragraph above Fig 5). 

5. In the case of images shooting from the same spot but different cameras as Leica Pegasus 2 and  Ladybug 5+, the problem of image quality and noise can be resolved by applying advanced filtering techniques. This matter should be noted and values research works to be cited as "Mediated morphological filters"(2001). 

[Reply 5.] 

Thank you very much for pointing us to this work. We will definitely take it into consideration once we are aiming to improve the image quality.  

Maybe there was a slight misunderstanding, because this paper just wants to show the differences between 2 different commercial mobile mapping systems, concerning their data quality (LIDAR, imagery). The paper is not aiming to address how image data quality could be optimized in post-processing.  

Maybe it would be a good idea to have a follow up paper, discussing different approaches of how to increase image quality. But this will be a future paper’s topic. 

This explanation is added to the end of section 2.3.3. The suggested reference ("Mediated morphological filters"(2001)) was cited in the revised paper. 

6. What exactly Figure 9 indicates? make it clear in detail. 

[Reply 6.]

Figure 9 demonstrates the differences in point scan density between the Z+F 9012 Laser scanner mounted on the Leica System and the VQ-450 on the Riegl. In both figures, the same roundabout exit arm scanned with the Z+F (top) and the Riegl (bottom) is shown. We've changed the picture to focus on the essentials: Only the point cloud scan without vector data is now depicted. We've now updated the caption with a description of what kind of scene can be seen and also added details:  

When comparing the ground level scan pattern of both systems, the better point density of the Leica system is demonstrated by a denser fishnet shape. Thus surface properties such as slope and lateral profile as well as cracks or holes can be identified faster. 

Caption:  

Comparison between the Z+F 9012 (top) and the Riegl VQ-450 (bottom) while scanning ground level of an exit arm in a roundabout. The reduced point density of the Riegl compared to the Z+F scanner is clearly visible. Both systems are dual profiler setups. 

7. Figures 12, and 13 should be merged. 

[Reply 7.]  

The pics have been merged according to your comment. 

8. In the introduction, one part should be added to discuss the role of IoT sensors in this technology, as IoT has a key role in the monitoring of vehicles as in the following research works its role has been given for the case of agriculture vehicles: "Economic data analytic AI technique on IoT edge devices for health monitoring of agriculture machines", "Economic IoT strategy: the future technology for health monitoring and diagnostic of agriculture vehicles".   

[Reply 8.]  

The IoT part has been added with the requested citations. 

Reviewer 2 Report

This paper presents the methodology and some results of an outdoor test campaign which targets automated driving scenarios. The results presented are highly relevant with current activities in this domain and the paper addresses important intersect for the readers. Nevertheless, I have some comments for the authors.

In overall, the paper seems quite too long. Especially, as a reader, I felt redundancy in some paragraphs of section 2. For example:

  • In $2.2.1 at line 182, one can read “The digital twin includes the road surface and any traffic infrastructure, as well as the road topology and environment models (topography).”
  • In $2.2.2 at line 244, one can read “It requires a detailed highly accurate digital twin about the environment with elements of surfaces, signals, traffic signs, tunnels, buildings etc.”

These two sentences could probably be summarized in only one. Thus, I would suggest to the authors to reshape the organization of this section 2.2 to clearly separate the concept of HD map and its challenges with the contributions provided by the two systems used during the measurement campaign.

A similar comment can apply to section 2.4 as well, although I appreciated that the authors provided detailed setup of the different vehicles. I would recommend verifying the different paragraphs to avoid unnecessary redundancy,

In addition, adding a general overview with the different systems at the beginning of Section 2 would help the reader to go through the different paragraphs of this section.

In section 3, the authors present measurement results of 5G technology. Although this section is very instructive, it is not clear if the results have been obtained during this measurement campaign or during another experimentation. The relation between this section and the other sections (especially section 2 and 4) of the paper is not clearly established and should be improved. Some questions that come to my mind are: Which participants have 5G capabilities? In which kind of scenarios is the 5G technology used?

In section 3.5, the authors detail their measurement method with two scenarios. To justify these scenarios, the authors refer to one of their previous work (line 602). As the paper should be self-content, could you please provide some motivation behind these two scenarios?

At line 609, the authors write “As expected the median values increase as the car speed increases (Figure 25)”. Only figures 23 and 24 show the results w.r.t car speed. There is probably a typo here. Also, increase of the median value of RTT is not present in Figure 23. Could you please explain why different behavior are observed between Figure 23 and Figure 24?

In section 4, the authors provide some of their measurement results. Between lines 661 and 663, they explain that 41 scenarios have been executed, making 137 test runs for three days. Could you please give more details on these scenarios? Although I understand 41 scenarios is a lot, there is probably a way to classify these scenarios in different categories according to their features (HD Map, Automated Vehicles, Infrastructure perception, 5G…). In my opinion, giving more information on the tested cases and their potential application to automated driving use cases would be a major added value to support such measurement campaign.

Conclusion do not mention test activities on 5G. Could you explain why this does not appear?

Finally, I found some small typos in the paper:

  • Line 155 “there” -> To be removed
  • Line 378, “relayed” -> relied
  • Line 681, “form” -> from
  • Line 803, “grater” -> greater

Please check to finalize the paper.

Author Response

Thank you for the thorough review and the valuable comments. 

We addressed all requested points and the corresponding corrections/extensions are emphasized by red color in the revised manuscript. 

1. In overall, the paper seems quite too long. Especially, as a reader, I felt redundancy in some paragraphs of section 2. For example: 

In $2.2.1 at line 182, one can read “The digital twin includes the road surface and any traffic infrastructure, as well as the road topology and environment models (topography).” 

In $2.2.2 at line 244, one can read “It requires a detailed highly accurate digital twin about the environment with elements of surfaces, signals, traffic signs, tunnels, buildings etc.” 

These two sentences could probably be summarized in only one. Thus, I would suggest to the authors to reshape the organization of this section 2.2 to clearly separate the concept of HD map and its challenges with the contributions provided by the two systems used during the measurement campaign. 

[Reply 1.] 

Authors tried to do their best to shorten the paper. However, other reviewers’ comments indicated to extend other sections and write additional parts.  

The mentioned redundancies have been removed and reorganized Section 2.2 to clearly separate the concept of HD map and its challenges with the contributions. 

2. A similar comment can apply to section 2.4 as well, although I appreciated that the authors provided detailed setup of the different vehicles. I would recommend verifying the different paragraphs to avoid unnecessary redundancy. 

[Reply 2.]  

Section 2.4 provides information about infrastructure sensors. Section 2.5 provides detailed setup information of the different vehicles which is needed due to the different participants. We also carried out a redundancy check. 

3. In addition, adding a general overview with the different systems at the beginning of Section 2 would help the reader to go through the different paragraphs of this section. 

[Reply 3.]  

Thank you for your suggestion, a brief general overview was added to the beginning of Section 2. 

4. In section 3, the authors present measurement results of 5G technology. Although this section is very instructive, it is not clear if the results have been obtained during this measurement campaign or during another experimentation. The relation between this section and the other sections (especially section 2 and 4) of the paper is not clearly established and should be improved. Some questions that come to my mind are: Which participants have 5G capabilities? In which kind of scenarios is the 5G technology used? 

[Reply 4.] 

Thank you for this comment. Department of Telecommunications and Media Informatics has a good relationship with on big Mobile Network Operator (MNO) in Hungary. This MNO provided the 5G coverage on the area of the M86 motorway during the measurement campaign.   

The end-devices during the measurements were placed in the M2 type BMW, but independent measurement results were also generated in static circumstances. 

5. In section 3.5, the authors detail their measurement method with two scenarios. To justify these scenarios, the authors refer to one of their previous work (line 602). As the paper should be self-content, could you please provide some motivation behind these two scenarios?  

[Reply 5.]  

Thanks for this valuable comment. In general, there can be several technical parameters that can affect network transmission. During the measurement campaign, the most fundamental parameters were examined, such as packet size, Inter-Arrival-Time, Round-Trip-Time and vehicle speed. The “const” scenario was created to serve as a reference measurement method, where IAT and packet size are fixed only the vehicle speed is the variable. An arbitrarily chosen mixture of these parameters was performed based on previous 5G measurements in the other scenario. The purpose of this measurement scenario is to offer a basic understanding of a more "real" traffic scenario where there are multiple applications and traffic sources. 

6. At line 609, the authors write “As expected the median values increase as the car speed increases (Figure 25)”. Only figures 23 and 24 show the results w.r.t car speed. There is probably a typo here. Also, increase of the median value of RTT is not present in Figure 23.  

[Reply 6.] 

This is a valid comment. Thank you. We have corrected this typo. 

7. Could you please explain why different behavior are observed between Figure 23 and Figure 24? 

[Reply 7.] 

The most probable explanation for this comment is the increasing inter-packet-gap (or IAT) between the packets. In the case of higher IAT, after a transmission, the radio bearer connection releases. Before the next packet, the radio bearer connection has to be set up again, which takes some milliseconds. However, this topic needs further examination. This explanation is also added to the paper. 

8. In section 4, the authors provide some of their measurement results. Between lines 661 and 663, they explain that 41 scenarios have been executed, making 137 test runs for three days. Could you please give more details on these scenarios? Although I understand 41 scenarios is a lot, there is probably a way to classify these scenarios in different categories according to their features (HD Map, Automated Vehicles, Infrastructure perception, 5G…). In my opinion, giving more information on the tested cases and their potential application to automated driving use cases would be a major added value to support such measurement campaign. 

[Reply 8.]  

Small description about the scenarios has been added. The description classifies the scenarios into five different categories based on their main features.  

9. Conclusion do not mention test activities on 5G. Could you explain why this does not appear? 

[Reply 9.]  

Thank you for the comment. It was missing. Conclusion has been extended accordingly. 

10. Finally, I found some small typos in the paper: 

- Line 155 “there” -> To be removed  

- Line 378, “relayed” -> relied  

- Line 681, “form” -> from  

- Line 803, “grater” -> greater  

[Reply 10.]  

The paper has been thoroughly revised and the typos have been removed.  

Reviewer 3 Report

The manuscript presents a measurement campaign on real motorway stretches using both vehicle and infrastructure sensors.

The presented work is comprehensive, with a good text organization and quality. Thus, this paper is appropriate for the SENSORS journal, and I think the audience will welcome the present contribution. Congratulations to team for their nice paper.

Author Response

Thank you for your review and positive opinion. 

English language check was carried out. Additionlly, we addressed the points requested by other reviewers and the corresponding corrections/extensions are emphasized by red color in the revised manuscript

Reviewer 4 Report

This is well prepared technical report from the motorway measurement campaign. The scientific content of the manuscript is low and limited to short description of the data analysis and processing. In order to improve the scientific soundness of the paper the following improvement could be suggested:

1) Include into the paper data processing and analysis flow and describe the details of the corresponding elements;

2) Describe test scenarios that were conducted during measurement campaign.

Author Response

Thank you for the thorough review and the valuable comments. 

We addressed all requested points and the corresponding corrections/extensions are emphasized by red color in the revised manuscript. 

1. Include into the paper data processing and analysis flow and describe the details of the corresponding elements; 

[Reply 1.]  

A figure is added for each example application, describing the data process method and detailing each step. 

2. Describe test scenarios that were conducted during measurement campaign. 

[Reply 2.]  

Small description about the scenarios has been added in section 4. The description classifies the scenarios into five different categories based on their main features. However, one of the reviewers stated that our article is too long, so the description must be kept short. 

Round 2

Reviewer 1 Report

As the authors have revised the paper well and followed the reviewers' comments in detail, the work is qualified for publication.

Author Response

Thank you for your valuable review work.

Reviewer 2 Report

The authors have adressed most of my comments for the previous review.

I still have some minor questions.

At lines 192 to 197, the authors claim that UHD maps for testing nd validation have absolute accuracy of +/- 2cm and commercial HD maps have accuracy of +/- 20cm. Could you give some references that support these figures?

At line 629, the authors write : "the outlier values are more widespread as the car speed increases (Figure 23)" I doubt there is some typo here as Figure 23 shows the RTT versus the packet size. Could you clarify this paragraph?

Author Response

Thanks you for your comments which really make our work better. We addressed the two remaining comments as follows.

1) We added a supporting reference to this part: 

ABI Research and HERE Technologies, “THE FUTURE OF MAPS: TECHNOLOGIES, PROCESSES, AND ECOSYSTEM", 2018. https://www.here.com/sites/g/files/odxslz166/files/2019-01/THE%20FUTURE%20OF%20MAPS.pdf

2) Thanks for the comment. We corrected this mistake as the references to Figures 23 and 24 were interchanged in the source file.

Reviewer 4 Report

I am satisfied with the improvements introduced to the manuscript. Thank you.

Author Response

(The authors gave the same response as above.)
